# Graph Constrained Reinforcement Learning for Natural Language Action Spaces

**Prithviraj Ammanabrolu**
Georgia Institute of Technology
raj.ammanabrolu@gatech.edu

**Matthew Hausknecht**
Microsoft Research
matthew.hausknecht@microsoft.com

## Abstract

Interactive Fiction games are text-based simulations in which an agent interacts with the world purely through natural language. They are ideal environments for studying how to extend reinforcement learning agents to meet the challenges of natural language understanding, partial observability, and action generation in combinatorially-large text-based action spaces. We present KG-A2C[1], an agent that builds a dynamic knowledge graph while exploring and generates actions using a template-based action space. We contend that the dual uses of the knowledge graph to reason about game state and to constrain natural language generation are the keys to scalable exploration of combinatorially large natural language actions. Results across a wide variety of IF games show that KG-A2C outperforms current IF agents despite the exponential increase in action space size.

## 1 Introduction

Natural language communication has long been considered a defining characteristic of human intelligence. We are motivated by the question of how learning agents can understand and generate contextually relevant natural language in service of achieving a goal. In pursuit of this objective we study Interactive Fiction (IF) games, or text-adventures: simulations in which an agent interacts with the world purely through natural language—"seeing" and "talking" to the world using textual descriptions and commands. To progress in these games, an agent must generate natural language actions that are coherent, contextually relevant, and able to effect the desired change in the world.

Complicating the problem of generating contextually relevant language in these games is the issue of *partial observability*: the fact that the agent never has access to the true underlying world state. IF games are structured as puzzles and often consist of an complex, interconnected web of distinct locations, objects, and characters. The agent needs to thus reason about the complexities of such a world solely through the textual descriptions that it receives, descriptions that are often incomplete. Further, an agent must be able to perform *commonsense reasoning*—IF games assume that human players possess prior commonsense and thematic knowledge—e.g. knowing that swords can kill trolls or that trolls live in dark places. Knowledge graphs provide us with an intuitive way of representing these partially observable worlds. Prior works have shown how using knowledge graphs aid in the twin issues of *partial observability* (Ammanabrolu & Riedl, 2019a) and *commonsense reasoning* (Ammanabrolu & Riedl, 2019b), but do not use them in the context of generating natural language.

To gain a sense for the challenges surrounding natural language generation, we need to first understand how large this space really is. In order to solve solve a popular IF game such as *Zork1* it's necessary to generate actions consisting of up to five-words from a relatively modest vocabulary of 697 words recognized by Zork's parser. Even this modestly sized vocabulary leads to $\mathcal{O}(697^5) = 1.64 \times 10^{14}$ possible actions at every step—a dauntingly-large *combinatorially-sized action space* for a learning agent to explore. In order to reduce the size of this space while maintaining expressiveness, Hausknecht et al. (2019a) propose the use of template-actions in which the agent first selects a template (e.g. [put] ___[in]___) then fills in the blanks using vocabulary words. There are 237 templates in *Zork1*, each with up to two blanks, yielding a template-action space of size

---

[1]Code available at https://github.com/rajammanabrolu/KG-A2C

$\mathcal{O}(237 \times 697^2) = 1.15 \times 10^8$. This space is six orders of magnitude smaller than the word-based space, but still six orders of magnitude larger than the action spaces used by previous text-based agents (Narasimhan et al., 2015; Zahavy et al., 2018). We demonstrate how these templates provide the structure required to further constrain our action space via our knowledge graph—and make the argument that the *combination* of these approaches allows us to generate meaningful natural language commands.

Our contributions are as follows: We introduce an novel agent that utilizes both a knowledge graph based state space and template based action space and show how to train such an agent. We then conduct an empirical study evaluating our agent across a diverse set of IF games followed by an ablation analysis studying the effectiveness of various components of our algorithm as well as its overall generalizability. Remarkably we show that our agent achieves state-of-the-art performance on a large proportion of the games despite the exponential increase in action space size.

## 2 RELATED WORK

We examine prior work in three broad categories: text-based game playing agents and frameworks as well as knowledge graphs used for natural language generation and game playing agents.

LSTM-DQN (Narasimhan et al., 2015), considers *verb-noun* actions up to two-words in length. Separate Q-Value estimates are produced for each possible verb and object, and the action consists of pairing the maximally valued verb combined with the maximally valued object. The DRRN algorithm for choice-based games (He et al., 2016; Zelinka, 2018) estimates Q-Values for a particular action from a particular state. Fulda et al. (2017) use Word2Vec (Mikolov et al., 2013) to aid in extracting affordances for items in these games and use this information to produce relevant action verbs. Zahavy et al. (2018) reduce the combinatorially-sized action space into a discrete form using a walkthrough of the game and introduce the Action Elimination DQN, which learns to eliminate actions unlikely to cause a world change.

Côté et al. (2018) introduce TextWorld, a framework for procedurally generating parser-based games, allowing a user to control the difficulty of a generated game. Yuan et al. (2019) introduce the concept of interactive question-answering in the form of QAit—modeling QA tasks in TextWorld. Urbanek et al. (2019) introduce Light, a dataset of crowdsourced text-adventure game dialogs focusing on giving collaborative agents the ability to generate contextually relevant dialog and emotes. Hausknecht et al. (2019a) have open-sourced Jericho[2], an optimized interface for playing human-made IF games—formalizing this task. They further provide a comparative study of various types of agents on their set of games, testing the performance of heuristic based agents such as NAIL (Hausknecht et al., 2019b) and various reinforcement learning agents are benchmarked. We use Jericho and the tools that it provides to develop our agents.

Knowledge graphs have been shown to be useful representations for a variety of tasks surrounding natural language generation and interactive fiction. Ghazvininejad et al. (2017) and Guan et al. (2018) effectively use knowledge graph representations to improve neural conversational and story ending prediction models respectively. Ammanabrolu et al. (2019) explore procedural content generation in text-adventure games—looking at constructing a quest for a given game world, and use knowledge graphs to ground generative systems trained to produce quest content. From the perspective of text-game playing agent and most in line with the spirit of our work, Ammanabrolu & Riedl (2019a) present the Knowledge Graph DQN or KG-DQN, an approach where a knowledge graph built during exploration is used as a state representation for a deep reinforcement learning based agent. Ammanabrolu & Riedl (2019b) further expand on this work, exploring methods of transferring control policies in text-games, using knowledge graphs to seed an agent with useful commonsense knowledge and to transfer knowledge between different games within a domain. Both of these works, however, identify a discrete set of actions required to play the game beforehand and so do not fully tackle the issue of the combinatorial action space.

---

[2]https://github.com/microsoft/jericho

## 3 STATE AND ACTION SPACES

Formally, IF games are partially observable Markov decision processes (POMDP), represented as a 7-tuple of $\langle S, T, A, \Omega, O, R, \gamma \rangle$ representing the set of environment states, mostly deterministic conditional transition probabilities between states, the vocabulary or words used to compose text commands, observations returned by the game, observation conditional probabilities, reward function, and the discount factor respectively (Côté et al., 2018; Hausknecht et al., 2019a). To deal with the resulting twin challenges of partial observability and combinatorial actions, we use a knowledge graph based state space and a template-based action space—each described in detail below.

**Knowledge Graph State Space.** Building on Ammanabrolu & Riedl (2019a), we use a knowledge graph as a state representation that is learnt during exploration. The knowledge graph is stored as a set of 3-tuples of $\langle subject, relation, object \rangle$. These triples are extracted from the observations using Stanford's Open Information Extraction (OpenIE) (Angeli et al., 2015). Human-made IF games often contain relatively complex semi-structured information that OpenIE is not designed to parse and so we add additional rules to ensure that we are parsing the relevant information.

Updated after every action, the knowledge graph helps the agent form a map of the world that it is exploring, in addition to retaining information that it has learned such as the affordances associated with an object, the properties of a character, current inventory, etc. Nodes relating to such information are shown on the basis of their relation to the agent which is presented on the graph using a "you" node (see example in Fig. 2a).

Ammanabrolu & Riedl (2019a) build a knowledge graph in a similar manner but restrict themselves to a single domain. In contrast, we test our methods on a much more diverse set of games defined in the Jericho framework (Hausknecht et al., 2019a). These games are each structured differently—covering a wider variety of genres—and so to be able to extract the same information from all of them in a general manner, we relax many of the rules found in Ammanabrolu & Riedl (2019a). To aid in the generalizability of graph building, we introduce the concept of *interactive objects*—items that an agent is able to directly interact with in the surrounding environment. These items are directly linked to the "you" node, indicating that the agent can interact with them, and the node for the current room, showing their relative position. All other triples built from the graph are extracted by OpenIE. Further details regarding knowledge graph updates are found in Appendix B.1 An example of a graph built using these rules is seen in Fig. 2a.

**Template Action Space.** Templates are subroutines used by the game's parser to interpret the player's action. They consist of interchangeable verbs phrases ($VP$) optionally followed by prepositional phrases ($VP\ PP$), e.g. ($[carry/hold/take]$ __) and ($[drop/throw/discard/put]$ __ $[at/against/on/onto]$ __), where the verbs and prepositions within [.] are aliases. As shown in Figure 2b, actions may be constructed from templates by filling in the template's blanks using words in the game's vocabulary. Templates and vocabulary words are programmatically accessible through the Jericho framework and are thus available for every IF game. Further details about how we prioritize interchangeable verbs and prepositions are available in Appendix B.2.

## 4 KNOWLEDGE GRAPH ADVANTAGE ACTOR CRITIC

Combining the knowledge-graph state space with the template action space, Knowledge Graph Advantage Actor Critic or KG-A2C, is an on-policy reinforcement learning agent that collects experience from many parallel environments. We first discuss the architecture of KG-A2C, then detail the training algorithm. As seen in Fig. 1, KG-A2C's architecture can broadly be described in terms of encoding a state representation and then using this encoded representation to decode an action. We describe each of these processes below.

**Input Representation.** The input representation network is broadly divided into three parts: an observation encoder, a score encoder, and the knowledge graph. At every step an observation consisting of several components is received: $o_t = (o_{t_{desc}}, o_{t_{game}}, o_{t_{inv}}, a_{t-1})$ corresponding to the room description, game feedback, inventory, and previous action, and total score $R_t$. The room description $o_{t_{desc}}$ is a textual description of the agent's location, obtained by executing the command "look." The game feedback $o_{t_{game}}$ is the simulators response to the agent's previous action and con-

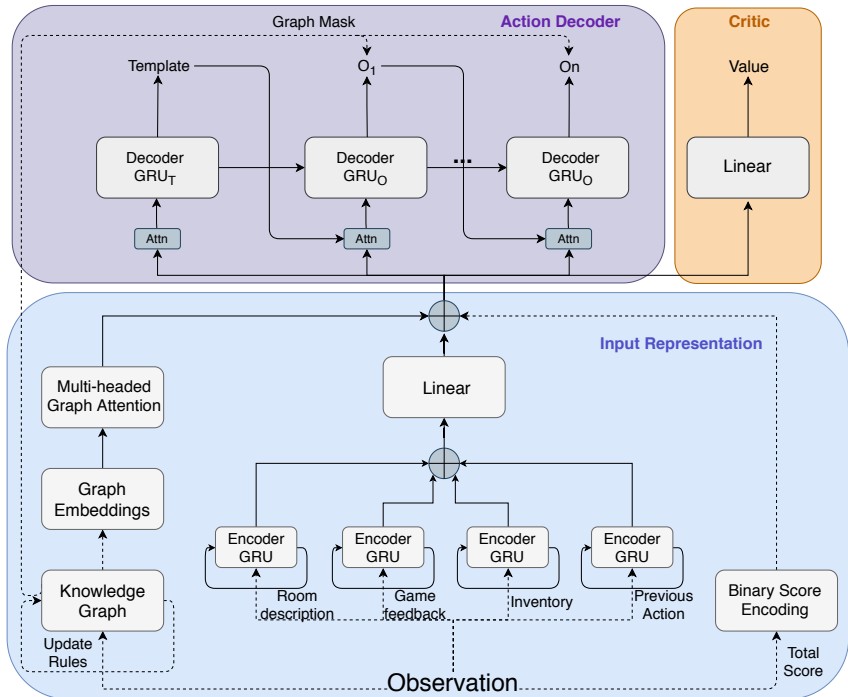

Figure 1: The full KG-A2C architecture. Solid lines represent computation flow along which the gradient can be back-propagated.

sists of narrative and flavor text. The inventory $o_{t_{inv}}$ and previous action $a_{t-1}$ components inform the agent about the contents of its inventory and the last action taken respectively.

The observation encoder processes each component of $o_t$ using a separate GRU encoder. As we are not given the vocabulary that $o_t$ is comprised of, we use subword tokenization—specifically using the unigram subword tokenization method described in Kudo & Richardson (2018). This method predicts the most likely sequence of subword tokens for a given input using a unigram language model which, in our case, is trained on a dataset of human playthroughs of IF games[3] and contains a total vocabulary of size 8000. For each of the GRUs, we pass in the final hidden state of the GRU at step $t-1$ to initialize the hidden state at step $t$. We concatenate each of the encoded components and use a linear layer to combine them into the final encoded observation $\mathbf{o}_t$.

At each step, we update our knowledge graph $G_t$ using $o_t$ as described in Sec. 3 and it is then embedded into a single vector $\mathbf{g_t}$. Following Ammanabrolu & Riedl (2019a) we use Graph Attention networks or GATs (Veličković et al., 2018) with an attention mechanism similar to that described in Bahdanau et al. (2014). Node features are computed as $H = \{\mathbf{h_1}, \mathbf{h_2}, \ldots, \mathbf{h_N}\}, \mathbf{h_i} \in \mathbb{R}^F$, where $N$ is the number of nodes and $F$ the number of features in each node, consist of the average subword embeddings of the entity and of the relations for all incoming edges using our unigram language model. Self-attention is then used after a learnable linear transformation $W \in \mathbb{R}^{2F \times F}$ applied to all the node features. Attention coefficients $\alpha_{ij}$ are then computed by softmaxing $k \in \mathcal{N}$ with $\mathcal{N}$ being the neighborhood in which we compute the attention coefficients and consists of all edges in $G_t$.

$$e_{ij} = LeakyReLU(\mathbf{p} \cdot W(\mathbf{h_i} \oplus \mathbf{h_j})) \tag{1}$$

$$\alpha_{ij} = \frac{exp(e_{ij})}{\sum_{k \in \mathcal{N}} exp(e_{ik})} \tag{2}$$

---

[3] http://www.allthingsjacq.com/interactive_fiction.html#clubfloyd

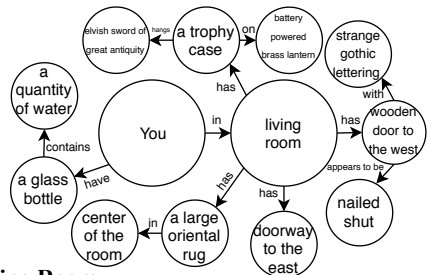

**Living Room**
You are in the living room. There is a doorway to the east, a **wooden door** with **strange gothic lettering** to the west, which appears to be nailed shut, a **trophy case**, and a **large oriental rug** in the center of the room. Above the trophy case hangs an elvish **sword** of great **antiquity**. A battery-powered brass **lantern** is on the trophy case.
You are carrying:
  A glass **bottle**
  The glass bottle contains:
    A quantity of water.

(a) The extracted knowledge graph for the corresponding state. Bolded words in the observation indicate *interactive objects*.

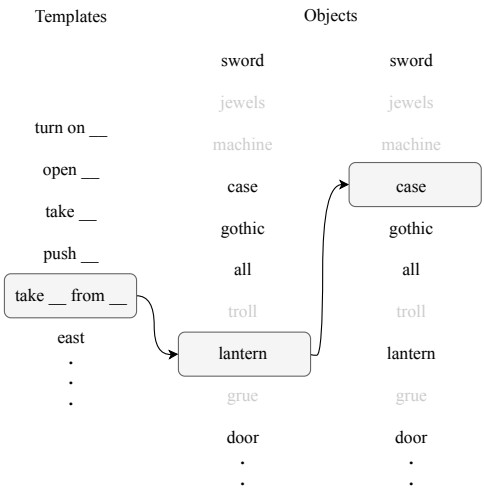

(b) Visualization of the action decoding process using templates and objects. Objects consist of the entire game input vocabulary. Greyed out words indicate objects masked out by the knowledge graph.

Figure 2: An overall example of the knowledge graph building and subsequent action decoding process for a given state in *Zork1*, illustrating the use of *interactive objects* and the *graph mask*.

where $\mathbf{p} \in \mathbb{R}^{2F}$ is a learnable parameter. The final knowledge graph embedding vector $\mathbf{g_t}$ is computed as:

$$\mathbf{g_t} = f(W_g(\bigoplus_{k=1}^{K} \sigma(\sum_{j \in \mathcal{N}} \alpha_{ij}^{(k)} \mathbf{W}^{(k)} \mathbf{h}_j)) + b_g) \tag{3}$$

where $k$ refers to the parameters of the $k^{th}$ independent attention mechanism, $W_g$ and $b_g$ the weights and biases of the output linear layer, and $\bigoplus$ represents concatenation. The final component of state embedding vector is a binary encoding $\mathbf{c}_t$ of the total score obtained so far in the game—giving the agent a sense for how far it has progressed in the game even when it is not collecting reward. The state embedding vector is then calculated as $\mathbf{s_t} = \mathbf{g_t} \oplus \mathbf{o_t} \oplus \mathbf{c_t}$.

**Action Decoder.** The state embedding vector $\mathbf{s_t}$ is then used to sequentially construct an action by first predicting a template and then picking the objects to fill into the template using a series of Decoder GRUs. This gives rise to a template policy $\pi_{\mathbb{T}}$ and a policy for each object $\pi_{\mathbb{O}_i}$. Architecture wise, at every decoding step all previously predicted parts of the action are encoded and passed along with $\mathbf{s}_t$ through an attention layer which learns to attend over these representations—conditioning every predicted object on all the previously predicted objects and template. All the object decoder GRUs share parameters while the template decoder GRU$_T$ remains separate.

To effectively constrain the space of template-actions, we introduce the concept of a *graph mask*, leveraging our knowledge graph at that timestep $G_t$ to streamline the object decoding process. Formally, the *graph mask* $m_t = \{o : o \in G_t \wedge o \in V\}$, consists of all the entities found within the knowledge graph $G_t$ and vocabulary $V$ and is applied to the outputs of the object decoder GRUs—restricting them to predict objects in the mask. Generally, in an IF game, it is impossible to interact with an object that you never seen or that are not in your inventory and so the mask lets us explore the action space more efficiently. To account for cases where this assumption does not hold, i.e. when an object that the agent has never interacted with before must be referenced in order to progress in the game, we randomly add objects $o \in V$ to $m_t$ with a probability $p_m$. An example of the graph-constrained action decoding process is illustrated in Fig. 2b.

### 4.1 TRAINING

We adapt the Advantage Actor Critic (A2C) method (Mnih et al., 2016) to train our network, using multiple workers to gather experiences from the simulator, making several significant changes along the way—as described below.

**Valid Actions.** Using a template-action space there are millions of possible actions at each step. Most of these actions do not make sense, are ungrammatical, etc. and an even fewer number of them actually cause the agent effect change in the world. Without any sense for which actions present valid interactions with the world, the combinatorial action space becomes prohibitively large for effective exploration.

We thus use the concept of *valid actions*, actions that can change the world in a particular state. These actions can usually be recognized through the game feedback, with responses like "Nothing happens" or "That phrase is not recognized." In practice, we follow Hausknecht et al. (2019a) and use the valid action detection algorithm provided by Jericho. Formally, $Valid(s_t) = \{a_0, a_1...a_N\}$ and from this we can construct the corresponding set of *valid templates* $\mathcal{T}_{valid}(s_t) = \{\tau_0, \tau_1...\tau_N\}$. We further define a set of *valid objects* $\mathcal{O}_{valid}(s_t) = \{o_0, o_1...o_M\}$ which consists of all objects in the graph mask as defined in Sec. 4. This lets us introduce two cross-entropy loss terms to aid the action decoding process. The template loss given a particular state and current network parameters is applied to the decoder $\text{GRU}_T$. Similarly, the object loss is applied across the decoder $\text{GRU}_O$ and is calculated by summing cross-entropy loss from all the object decoding steps.

$$\mathcal{L}_{\mathbb{T}}(s_t, a_t; \theta_t) = \frac{1}{N} \sum_{i=1}^{N} (y_{\tau_i} log \pi_{\mathbb{T}}(\tau_i|s_t) + (1 - y_{\tau_i})(1 - log \pi_{\mathbb{T}}(\tau_i|s_t)) \tag{4}$$

$$\mathcal{L}_{\mathbb{O}}(s_t, a_t; \theta_t) = \sum_{j=1}^{n} \frac{1}{M} \sum_{i=1}^{M} (y_{o_i} log \pi_{\mathbb{O}_j}(o_i|s_t) + (1 - y_{o_i})(1 - log \pi_{\mathbb{O}_j}(o_i|s_t))) \tag{5}$$

$$y_{\tau_i} = \begin{cases} 1 & \tau_i \in \mathcal{T}_{valid}(s_t) \\ 0 & else \end{cases} \quad y_{o_i} = \begin{cases} 1 & o_i \in \mathcal{O}_{valid}(s_t) \\ 0 & else \end{cases}$$

**Updates.** A2C training starts with calculating the advantage of taking an action in a state $A(s_t, a_t)$, defined as the value of taking an action $Q(s_t, a_t)$ compared to the average value of taking all possible *valid actions* in that state $V(s_t)$:

$$A(s_t, a_t) = Q(s_t, a_t) - V(s_t) \tag{6}$$

$$Q(s_t, a_t) = \mathbb{E}[r_t + \gamma V(s_{t+1})] \tag{7}$$

$V(s_t)$ is predicted by the critic as shown in Fig. 1 and $r_t$ is the reward received at step $t$.

The action decoder or actor is then updated according to the gradient:

$$-\nabla_\theta (log \pi_{\mathbb{T}}(\tau|s_t; \theta_t) + \sum_{i=1}^{n} log \pi_{\mathbb{O}_i}(o_i|s_t, \tau, ..., o_{i-1}; \theta_t)) A(s_t, a_t) \tag{8}$$

updating the template policy $\pi_{\mathbb{T}}$ and object policies $\pi_{\mathbb{O}_i}$ based on the fact that each step in the action decoding process is conditioned on all the previously decoded portions. The critic is updated with respect to the gradient:

$$\frac{1}{2} \nabla_\theta (Q(s_t, a_t; \theta_t) - V(s_t; \theta_t))^2 \tag{9}$$

bringing the critic's prediction of the value of being in a state closer to its true underlying value. We further add an entropy loss over the valid actions, designed to prevent the agent from prematurely converging on a trajectory.

$$\mathcal{L}_{\mathbb{E}}(s_t, a_t; \theta_t) = \sum_{a \in V(s_t)} P(a|s_t) log P(a|s_t) \tag{10}$$

## 5 EXPERIMENTAL RESULTS

The KG-A2C is tested on a suite of Jericho supported games and is compared to strong, established baselines. Additionally, as encouraged by Hausknecht et al. (2019a), we present the set of handicaps used by our agents: (1) Jericho's ability to identify *valid actions* and (2) the Load, Save handicap in order to acquire $o_{t_{desc}}$ and $o_{t_{inv}}$ using the *look* and *inventory* commands without changing the game state. Hyperparameters are provided in Appendix C.

**Template DQN Baseline.** We compare KG-A2C against Template-DQN, a strong baseline also utilizing the template based action space. TDQN (Hausknecht et al., 2019a) is an extension of LSTM-DQN (Narasimhan et al., 2015) to template-based action spaces. This is accomplished using three output heads: one for estimating the Q-Values over templates $Q(s_t, u) \forall u \in \mathcal{T}$ and two for estimating Q-Values $Q(s_t, o_1), Q(s_t, o_2) \forall o_i \in \mathcal{O}$ over vocabulary to fill in the blanks of the template. The final executed action is constructed by greedily sampling from the predicted Q-values. Importantly, TDQN uses the same set of handicaps as KG-A2C allowing a fair comparison between these two algorithms.

Table 1 shows how KG-A2C fares across a diverse set of games supported by Jericho—testing the agent's ability to generalize to different genres, game structures, reward functions, and state-action spaces. KG-A2C matches or outperforms TDQN on 23 out of the 28 games that we test on. Our agent is thus shown to be capable of extracting a knowledge graph that can sufficiently constrain the template based action space to enable effective exploration in a broad range of games.

## 6 ABLATION STUDY

In order to understand the contributions of different components of KG-A2C's architecture, we ablate KG-A2C's knowledge graph, template-action space, and valid-action loss. These ablations are performed on *Zork1*[4] and result in the following agents:

**A2C** removes all components of KG-A2C's knowledge graph. In particular, the state embedding vector is now computed as $\mathbf{s_t} = \mathbf{o_t} \oplus \mathbf{c_t}$ and the *graph mask* is not used to constrain action decoding.

**KG-A2C-no-gat** remove's the Graph Attention network, but retains the graph masking components. The knowledge graph is still constructed as usual but the agent uses the same state embedding vector as A2C.

**KG-A2C-no-mask** ablates the graph mask for purposes of action decoding. The knowledge graph is constructed as usual and the agent retains graph attention.

On *Zork1* as shown in Figure 3, we observe similar asymptotic performance between the all of the ablations – all reach approximately 34 points. This level of performance corresponds to a local optima where the agent collects the

| Game | $|T|$ | $|V|$ | TDQN | KGA2C | MaxRew |
|---|---|---|---|---|---|
| 905 | 82 | 296 | 0 | 0 | 1 |
| acorncourt | 151 | 343 | **1.6** | 0.3 | 30 |
| advent[†] | 189 | 786 | 36 | 36 | 350 |
| adventureland | 156 | 398 | 0 | 0 | 100 |
| anchor | 260 | 2257 | 0 | 0 | 100 |
| awaken | 159 | 505 | 0 | 0 | 50 |
| balances | 156 | 452 | 4.8 | **10** | 51 |
| deephome | 173 | 760 | 1 | 1 | 300 |
| detective | 197 | 344 | 169 | **207.9** | 360 |
| dragon | 177 | 1049 | -5.3 | **0** | 25 |
| enchanter | 290 | 722 | 8.6 | **12.1** | 400 |
| inhumane | 141 | 409 | 0.7 | **3** | 300 |
| jewel | 161 | 657 | 0 | **1.8** | 90 |
| karn | 161 | 657 | **1.2** | 0 | 90 |
| library | 173 | 510 | 6.3 | **14.3** | 30 |
| ludicorp | 187 | 503 | 6 | **17.8** | 150 |
| moonlit | 166 | 669 | 0 | 0 | 1 |
| omniquest | 207 | 460 | **16.8** | 3 | 50 |
| pentari | 155 | 472 | 17.4 | **50.7** | 70 |
| snacktime | 201 | 468 | **9.7** | 0 | 50 |
| sorcerer | 288 | 1013 | 5 | **5.8** | 400 |
| spellbrkr | 333 | 844 | 18.7 | **21.3** | 600 |
| spirit | 169 | 1112 | 0.6 | **1.3** | 250 |
| temple | 175 | 622 | **7.9** | 7.6 | 35 |
| zenon | 149 | 401 | 0 | **3.9** | 350 |
| zork1 | 237 | 697 | 9.9 | **34** | 350 |
| zork3 | 214 | 564 | 0 | **.1** | 7 |
| ztuu | 186 | 607 | 4.9 | **9.2** | 100 |

Table 1: Raw scores comparing KG-A2C to TDQN across a wide set of games supported by Jericho. [†]Advent starts at a score of 36.

majority of available rewards without fighting the troll. Several other authors also report scores at this threshold (Jain et al., 2019; Zahavy et al., 2018). In terms of learning speed, the methods which have access to either the graph attention or the graph mask converge slightly faster than pure A2C which has neither.

---

[4]A map of Zork1 with annotated rewards can be found in Appendix D along with a transcript of KG-A2C playing this game.

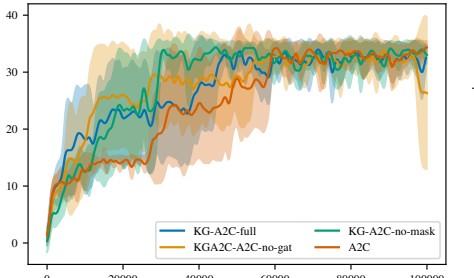

| Agent | Mask | GAT |
|---|---|---|
| A2C | | |
| KG-A2C-no-gat | ✓ | |
| KG-A2C-no-mask | | ✓ |
| KG-A2C-full | ✓ | ✓ |
| KG-A2C-unsup | ✓ | ✓ |

Figure 3: Ablation results on *Zork1*, averaged across 5 independent runs.

To further understand these differences we performed a larger study across the full set of games comparing KG-A2C-full with KG-A2C-no-mask. The results in Table 2 show KG-A2C-full outperforms KG-A2C-no-mask on 10 games and is outperformed by KG-A2C-no-mask on 6. From this larger study we thus conclude the graph mask and knowledge graph are broadly useful components.

We perform two final ablations to study the importance of the supervised valid-action loss and the template action space:

**KG-A2C-unsupervised** In order to understand the importance of training with valid-actions, KG-A2C-unsupervised is not allowed to access the list of *valid actions*—the valid-action-losses $\mathcal{L}_{\mathbb{T}}$ and $\mathcal{L}_{\mathbb{O}}$ are disabled and $\mathcal{L}_{\mathbb{E}}$ now based on the full action set. Thus, the agent must explore the template action space manually. KG-A2C-unsupervised, when trained for the same number of steps as all the other agents, fails to achieve any score. We can infer that the valid action auxiliary loss remains an important part of the overall algorithm, and access to the knowledge graph alone is not yet sufficient for removing this auxiliary loss.

**KG-A2C-seq** discards the template action space and instead decodes actions word by word up to a maximum of four words. A supervised cross-entropy-based valid action loss $\mathcal{L}_{\text{Valid}}$ is now calculated by selecting a random valid action $a_{t_{\text{valid}}} \in \text{Valid}(s_t)$ and using each token in it as a target label. As this action space is orders of magnitude larger than template actions, we use teacher-forcing to enable more effective exploration while training the agent—executing $a_{t_{\text{valid}}}$ with a probability $p_{\text{valid}} = 0.5$ and the decoded action otherwise. All other components remain the same as in the full KG-A2C.

KG-A2C-seq reaches a relatively low asymptotic performance of 8 points. This agent, using a action space consisting of the full vocabulary, performs significantly worse than the rest of the agents even when given the handicaps of teacher forcing and being allowed to train for significantly longer—indicating that the template based action space is also necessary for effective exploration.

# 7 CONCLUSION

Tabula rasa reinforcement learning offers an intuitive paradigm for exploring goal driven, contextually aware natural language generation. The sheer size of the natural language action space, however, has proven to be out of the reach of existing algorithms. In this paper we introduced KG-A2C, a novel learning agent that demonstrates the feasibility of scaling reinforcement learning towards natural language actions spaces with hundreds of millions of actions. The key insight to being able to efficiently explore such large spaces is the combination of a *knowledge-graph-based state space* and a *template-based action space*. The knowledge graph serves as a means for the agent to understand its surroundings, accumulate information about the game, and disambiguate similar textual observations while the template-based action space lends a measure of structure that enables us to exploit that same knowledge graph for language generation. Together they constrain the vast space of possible actions into the compact space of sensible ones. A suite of experiments across a diverse set of 28 human-made IF games shows wide improvement over TDQN, the current state-of-the-art template-based agent. Finally, an ablation study replicates state-of-the-art performance on *Zork1* even though KG-A2C is using an action space six orders of magnitude larger than previous agents—indicating the overall efficacy of our combined state-action space.

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

## A    ABLATION RESULTS

| Game | $|T|$ | $|V|$ | KGA2C-Full | KGA2C-unmasked | **MaxRew** |
|---|---|---|---|---|---|
| 905 | 82 | 296 | 0 | 0 | 1 |
| acorncourt | 151 | 343 | 0.3 | 0.3 | 30 |
| advent[†] | 189 | 786 | 36 | 36 | 350 |
| adventureland | 156 | 398 | 0 | 0 | 100 |
| anchor | 260 | 2257 | 0 | 0 | 100 |
| awaken | 159 | 505 | 0 | 0 | 50 |
| balances | 156 | 452 | 10 | 10 | 51 |
| deephome | 173 | 760 | 1 | **29.2** | 300 |
| detective | 197 | 344 | **207.9** | 141 | 360 |
| dragon | 177 | 1049 | **0** | -.2 | 25 |
| enchanter | 290 | 722 | **12.1** | 7.6 | 400 |
| inhumane | 141 | 409 | 3 | **10.2** | 300 |
| jewel | 161 | 657 | **1.8** | 1.3 | 90 |
| karn | 161 | 657 | 0 | 0 | 90 |
| library | 173 | 510 | **14.3** | 9.6 | 30 |
| ludicorp | 187 | 503 | 17.8 | **17.9** | 150 |
| moonlit | 166 | 669 | 0 | 0 | 1 |
| omniquest | 207 | 460 | 3 | **5.4** | 50 |
| pentari | 155 | 472 | **50.7** | 50.4 | 70 |
| snacktime | 201 | 468 | 0 | 0 | 50 |
| sorcerer | 288 | 1013 | 5.8 | **16.8** | 400 |
| spellbrkr | 333 | 844 | 21.3 | **30.1** | 600 |
| spirit | 169 | 1112 | 1.3 | 1.3 | 250 |
| temple | 175 | 622 | **7.6** | 6.4 | 35 |
| zenon | 149 | 401 | **3.9** | 3.1 | 350 |
| zork1 | 237 | 697 | **34** | 27 | 350 |
| zork3 | 214 | 564 | .1 | .1 | 7 |
| ztuu | 186 | 607 | **9.2** | 5 | 100 |

Table 2: Ablations

## B    IMPLEMENTATION DETAILS

### B.1    KNOWLEDGE GRAPH UPDATE RULES

Candidate interactive objects are identified by performing part-of-speech tagging on the current observation, identifying singular and proper nouns as well as adjectives, and are then filtered by checking if they can be examined using the command $examine\ OBJ$. Only the interactive objects not found in the inventory are linked to the node corresponding to the current room and the inventory items are linked to the "you" node. The only other rule applied uses the navigational actions performed by the agent to infer the relative positions of rooms, e.g. $\langle kitchen, down, cellar \rangle$ when the agent performs $go\ down$ when in the kitchen to move to the cellar.

### B.2    TEMPLATE PREPROCESSING

Templates are processed by selecting a single verb and preposition from the aliases. For the sake of agent explainability, we pick the verb and preposition that are most likely to be used by humans when playing IF games. This is done by assessing token frequencies from a dataset of human playthroughs such as those given in ClubFloyd[5]. This dataset consists of 425 unique play sessions and 273,469 state-action pairs. The examples given earlier, ($[carry/hold/take]$ __) and ($[drop/throw/discard/put]$ __ $[at/against/on/onto]$ __), would then be converted to $take$ __ and $put$ __ $on$ __.

---

[5]http://www.allthingsjacq.com/interactive_fiction.html#clubfloyd

# C  EXPERIMENT DETAILS

Episodes are terminated after 100 valid steps or game over/victory. Agents that decode invalid actions often wouldn't make it very far into the game, and so we only count valid-actions against the hundred step limit. All agents are trained individually on each game and then evaluated on that game. All A2C based agents are trained using data collected from 32 parallel environments. TDQN was trained using a single environment. Hyperparameters for all agents were tuned on the game of *Zork1* and held constant across all other games. Final reported scores are an average over 5 runs of each algorithm.

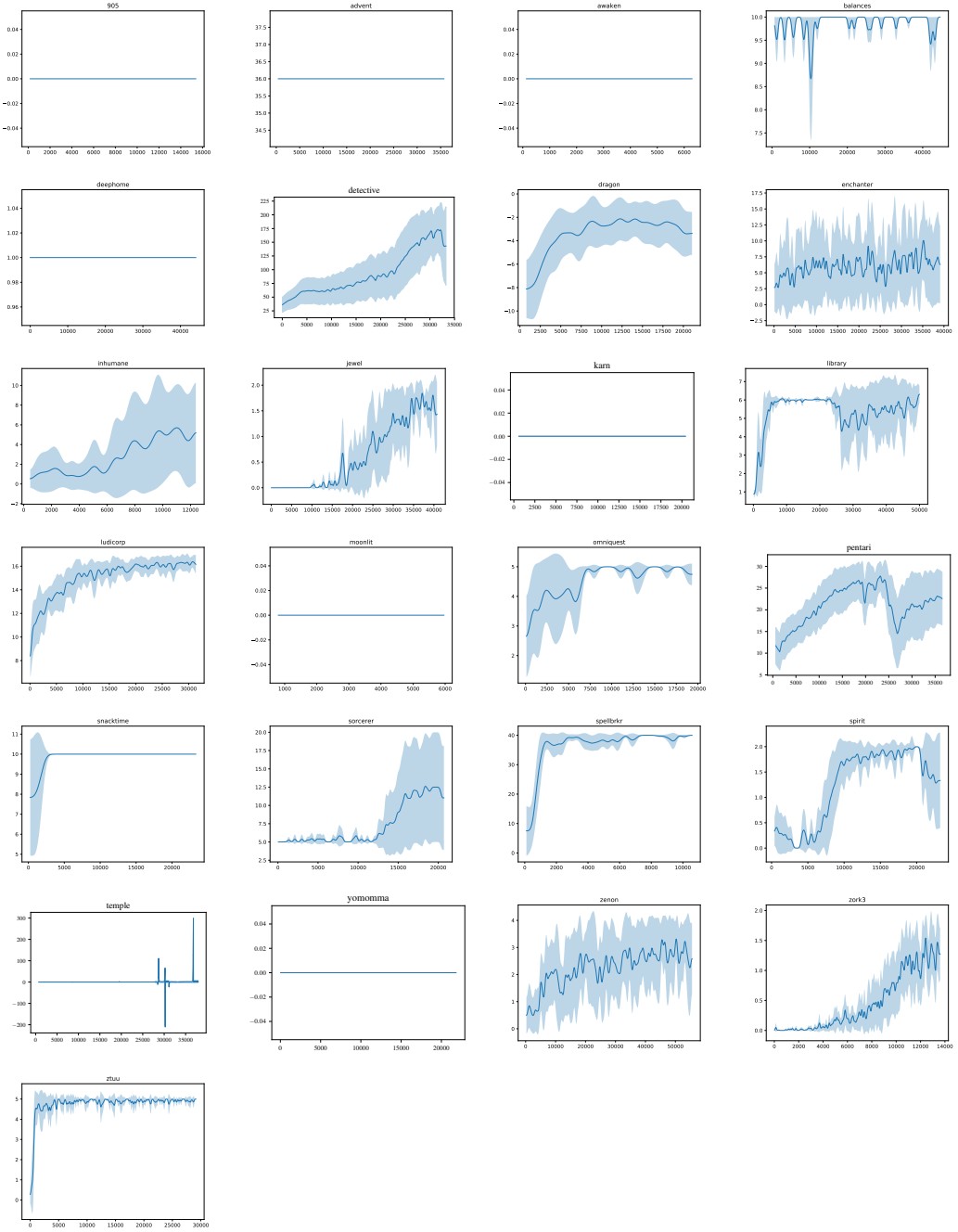

Figure 4: Learning curves for KGA2C-full. Shaded regions indicate standard deviations.

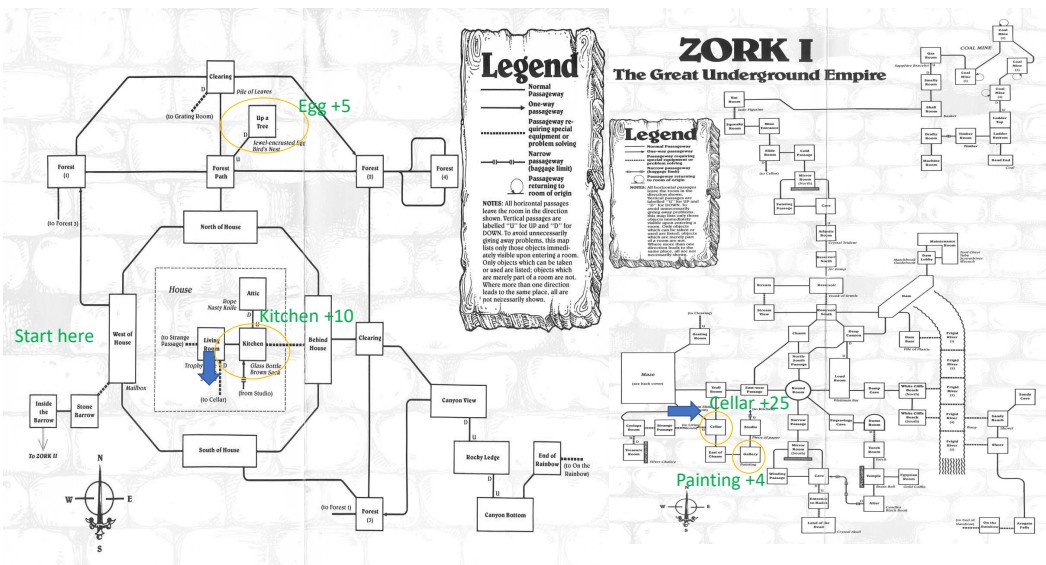

Figure 5: A map of the world of *Zork1* with some initial rewards annotated. The blue arrow indicates a connection between the left and right maps, corresponding to the overworld and the dungeon.

## D  ZORK1

*Zork1* was identified by Hausknecht et al. (2019a) to be one of the most difficult games in their suite and the subject of much prior work (Zahavy et al., 2018; Yin & May, 2019). *Zork1* is one of the earliest IF games and is a dungeon-crawler—a player must explore a vast labyrinth while fighting off enemies and complete puzzles in order to collect treasures. It features a relatively sparse reward for collecting a treasure or moving along the right path to one, and stochasticity in terms of random enemy movements.

To understand how humans progress in *Zork1*, a group of 10 human players—familiar with IF games—were asked to play *Zork1* for the first time (with no access to walkthroughs). Half of the players reached a game score of around 40 before dying to the first beatable NPC, a troll, mostly due to neglecting to collect a weapon to fight it with beforehand. Three of the remaining players died to hidden traps even before reaching this point, achieving scores between 5 and 15. The final two players made it significantly past the troll gaining scores of around 70.

The following transcript of KG-A2C playing *Zork1*, shows top predicted probabilities for templates and objects.

```
Obs: Desc: West of House You are standing in an open field west of a white house, with a boarded front door.
    There is a small mailbox here. Inv: You are emptyhanded. Feedback: West of House You are standing in an
    open field west of a white house, with a boarded front door. There is a small mailbox here.
Template probs: open OBJ: 0.339, east: 0.271, south: 0.215, west: 0.094, north: 0.031, go around OBJ: 0.013,
    blow out OBJ: 0.011, throw OBJ at OBJ: 0.01, northwest: 0.009
Object probs: window: 0.7, small: 0.299
Interactive objects: boarded, small, front, door, white, mailbox, house, all
Action: west
Score: 0
---
Obs: Desc: Forest This is a forest, with trees in all directions. To the east, there appears to be sunlight.
    Inv: You are emptyhanded. Feedback: Forest This is a forest, with trees in all directions. To the east,
    there appears to be sunlight.
Template probs: open OBJ: 0.49, south: 0.304, east: 0.116, west: 0.04, north: 0.016, go around OBJ: 0.011,
    throw OBJ at OBJ: 0.006, northwest: 0.006, blow out OBJ: 0.006
Object probs: window: 0.7, small: 0.3
Interactive objects: forest, all, trees
Action: east
Score: 0
---
Obs: Desc: Forest Path This is a path winding through a dimly lit forest. The path heads northsouth here. One
    particularly large tree with some low branches stands at the edge of the path. Inv: You are emptyhanded.
     Feedback: Forest Path This is a path winding through a dimly lit forest. The path heads northsouth here
    . One particularly large tree with some low branches stands at the edge of the path.
```

```
Template probs: up: 0.386, open OBJ: 0.358, east: 0.123, west: 0.06, north: 0.023, go around OBJ: 0.014, blow
    out OBJ: 0.01, northwest: 0.009, throw OBJ at OBJ: 0.008
Object probs: window: 0.701, small: 0.299
Interactive objects: tree, path, branches, forest, large, all
Action: up
Score: 0
---
Obs: Desc: Up a Tree You are about 10 feet above the ground nestled among some large branches. The nearest
    branch above you is above your reach. Beside you on the branch is a small birds nest. In the birds nest
    is a large egg encrusted with precious jewels, apparently scavenged by a childless songbird. The egg is
    covered with fine gold inlay, and ornamented in lapis lazuli and motherofpearl. Unlike most eggs, this
    one is hinged and closed with a delicate looking clasp. The egg appears extremely fragile. Inv: You are
    emptyhanded. Feedback: Up a Tree You are about 10 feet above the ground nestled among some large
    branches. The nearest branch above you is above your reach. Beside you on the branch is a small birds
    nest. In the birds nest is a large egg encrusted with precious jewels, apparently scavenged by a
    childless songbird. The egg is covered with fine gold inlay, and ornamented in lapis lazuli and
    motherofpearl. Unlike most eggs, this one is hinged and closed with a delicate looking clasp. The egg
    appears extremely fragile.
Template probs: west: 0.43, open OBJ: 0.18, go around OBJ: 0.074, east: 0.064, north: 0.063, south: 0.052,
    take OBJ: 0.041, put down OBJ: 0.023, down: 0.014, close OBJ: 0.009
Object probs: small: 0.213, window: 0.184, house: 0.116, egg: 0.068, white: 0.067, path: 0.062, branch: 0.042,
    kitche: 0.024, nest: 0.018, ground: 0.009
Interactive objects: nest, large, egg, ground, branches, all, branch, tree
Action: open nest
Score: 0
---
Obs: Desc: Up a Tree You are about 10 feet above the ground nestled among some large branches. The nearest
    branch above you is above your reach. Beside you on the branch is a small birds nest. In the birds nest
    is a large egg encrusted with precious jewels, apparently scavenged by a childless songbird. The egg is
    covered with fine gold inlay, and ornamented in lapis lazuli and motherofpearl. Unlike most eggs, this
    one is hinged and closed with a delicate looking clasp. The egg appears extremely fragile. You hear in
    the distance the chirping of a song bird. Inv: You are emptyhanded. You hear in the distance the
    chirping of a song bird. Feedback: It is already open.
Template probs: west: 0.347, open OBJ: 0.182, go around OBJ: 0.085, east: 0.077, north: 0.073, south: 0.058,
    take OBJ: 0.045, put down OBJ: 0.026, down: 0.018, close OBJ: 0.013
Object probs: small: 0.213, window: 0.183, house: 0.117, egg: 0.068, white: 0.067, path: 0.062, branch: 0.042,
    kitche: 0.024, nest: 0.018, ground: 0.009
Interactive objects: nest, large, egg, ground, branches, all, branch, tree
Action: take egg
Score: 0
---
Obs: Desc: Up a Tree You are about 10 feet above the ground nestled among some large branches. The nearest
    branch above you is above your reach. Beside you on the branch is a small birds nest. Inv: You are
    carrying A jewelencrusted egg Feedback: Taken.
Template probs: down: 0.417, open OBJ: 0.181, go around OBJ: 0.074, east: 0.065, north: 0.063, south: 0.053,
    take OBJ: 0.042, put down OBJ: 0.025, down: 0.014, close OBJ: 0.01
Object probs: small: 0.213, window: 0.184, house: 0.116, egg: 0.068, white: 0.067, path: 0.062, branch: 0.042,
    kitche: 0.024, nest: 0.018, ground: 0.009
Interactive objects: nest, large, egg, ground, branches, all, branch, tree
Action: down
Score: 5
---
Obs: Desc: Forest Path This is a path winding through a dimly lit forest. The path heads northsouth here. One
    particularly large tree with some low branches stands at the edge of the path. Inv: You are emptyhanded.
    Feedback: Forest Path
Template probs: south: 0.386, open OBJ: 0.358, east: 0.123, west: 0.06, north: 0.023, go around OBJ: 0.014,
    blow out OBJ: 0.01, northwest: 0.009, throw OBJ at OBJ: 0.008
Object probs: window: 0.701, small: 0.299
Interactive objects: tree, path, branches, forest, large, all
Action: south
Score: 5
---
Obs: Desc: North of House You are facing the north side of a white house. There is no door here, and all the
    windows are boarded up. To the north a narrow path winds through the trees. Inv: You are emptyhanded.
    Feedback: North of House You are facing the north side of a white house. There is no door here, and all
    the windows are boarded up. To the north a narrow path winds through the trees.
Template probs: west: 0.502, open OBJ: 0.177, west: 0.147, north: 0.094, south: 0.035, throw OBJ at OBJ:
    0.012, go around OBJ: 0.012, northwest: 0.01, blow out OBJ: 0.005
Object probs: window: 0.724, small: 0.243, trap: 0.014, kitche: 0.009, rug: 0.003
Interactive objects: path, windows, narrow, white, house, all
Action: west
Score: 5
---
Obs: Desc: West of House You are standing in an open field west of a white house, with a boarded front door.
    There is a small mailbox here. Inv: You are emptyhanded. Feedback:
Template probs: open OBJ: 0.536, south: 0.241, east: 0.121, west: 0.053, north: 0.015, go around OBJ: 0.011,
    throw OBJ at OBJ: 0.007, blow out OBJ: 0.006, northwest: 0.005
Object probs: window: 0.7, small: 0.3
Interactive objects: boarded, small, front, door, white, mailbox, house, all
Action: open white
Score: 5
---
Obs: Desc: West of House You are standing in an open field west of a white house, with a boarded front door.
    There is a small mailbox here. Inv: You are emptyhanded. Feedback: I cant see how to get in from here.
Template probs: open OBJ: 0.339, east: 0.271, south: 0.215, west: 0.094, north: 0.031, go around OBJ: 0.013,
    blow out OBJ: 0.011, throw OBJ at OBJ: 0.01, northwest: 0.009
Object probs: window: 0.7, small: 0.299
Interactive objects: boarded, small, front, door, white, mailbox, house, all
Action: west
Score: 5
---
```

```
Obs: Desc: Forest This is a forest, with trees in all directions. To the east, there appears to be sunlight.
     Inv: You are emptyhanded. Feedback: Forest This is a forest, with trees in all directions. To the east,
     there appears to be sunlight.
Template probs: open OBJ: 0.49, south: 0.304, east: 0.116, west: 0.04, north: 0.016, go around OBJ: 0.011,
     throw OBJ at OBJ: 0.006, northwest: 0.006, blow out OBJ: 0.006
Object probs: window: 0.7, small: 0.3
Interactive objects: forest, all, trees
Action: east
Score: 5
---
Obs: Desc: Forest Path This is a path winding through a dimly lit forest. The path heads northsouth here. One
     particularly large tree with some low branches stands at the edge of the path. Inv: You are emptyhanded.
     Feedback: Forest Path This is a path winding through a dimly lit forest. The path heads northsouth here
     . One particularly large tree with some low branches stands at the edge of the path.
Template probs: south: 0.386, open OBJ: 0.358, east: 0.123, west: 0.06, north: 0.023, go around OBJ: 0.014,
     blow out OBJ: 0.01, northwest: 0.009, throw OBJ at OBJ: 0.008
Object probs: window: 0.701, small: 0.298
Interactive objects: tree, path, branches, forest, large, all
Action: open all
Score: 5
---
Obs: Desc: Forest Path This is a path winding through a dimly lit forest. The path heads northsouth here. One
     particularly large tree with some low branches stands at the edge of the path. Inv: You are emptyhanded.
     Feedback: Its not clear what youre referring to.
Template probs: open OBJ: 0.331, south: 0.259, east: 0.24, west: 0.087, north: 0.031, go around OBJ: 0.013,
     blow out OBJ: 0.011, northwest: 0.01, throw OBJ at OBJ: 0.009
Object probs: window: 0.701, small: 0.299
Interactive objects: tree, path, branches, forest, large, all
Action: east
Score: 5
---
Obs: Desc: Forest This is a dimly lit forest, with large trees all around. Inv: You are emptyhanded. You hear
     in the distance the chirping of a song bird. Feedback: Forest This is a dimly lit forest, with large
     trees all around. You hear in the distance the chirping of a song bird.
Template probs: south: 0.38, open OBJ: 0.281, east: 0.174, west: 0.072, north: 0.034, go around OBJ: 0.015,
     blow out OBJ: 0.013, northwest: 0.012, throw OBJ at OBJ: 0.008
Object probs: window: 0.7, small: 0.299
Interactive objects: forest, large, all, trees
Action: south
Score: 5
---
Obs: Desc: Clearing You are in a small clearing in a well marked forest path that extends to the east and west
     . Inv: You are emptyhanded. Feedback: Clearing You are in a small clearing in a well marked forest path
     that extends to the east and west.
Template probs: open OBJ: 0.451, east: 0.364, west: 0.164, north: 0.008, throw OBJ at OBJ: 0.005
Object probs: window: 0.592, small: 0.405, kitche: 0.002
Interactive objects: forest, path, all
Action: east
Score: 5
---
Obs: Desc: Canyon View You are at the top of the Great Canyon on its west wall. From here there is a marvelous
      view of the canyon and parts of the Frigid River upstream. Across the canyon, the walls of the White
     Cliffs join the mighty ramparts of the Flathead Mountains to the east. Following the Canyon upstream to
     the north, Aragain Falls may be seen, complete with rainbow. The mighty Frigid River flows out from a
     great dark cavern. To the west and south can be seen an immense forest, stretching for miles around. A
     path leads northwest. It is possible to climb down into the canyon from here. Inv: You are emptyhanded.
     Feedback: Canyon View You are at the top of the Great Canyon on its west wall. From here there is a
     marvelous view of the canyon and parts of the Frigid River upstream. Across the canyon, the walls of the
      White Cliffs join the mighty ramparts of the Flathead Mountains to the east. Following the Canyon
     upstream to the north, Aragain Falls may be seen, complete with rainbow. The mighty Frigid River flows
     out from a great dark cavern. To the west and south can be seen an immense forest, stretching for miles
     around. A path leads northwest. It is possible to climb down into the canyon from here.
Template probs: west: 0.444, take OBJ: 0.204, open OBJ: 0.147, blow out OBJ: 0.118, turn on OBJ: 0.027, east:
     0.026, down: 0.009, go around OBJ: 0.005, push OBJ: 0.004
Object probs: window: 0.693, small: 0.307
Interactive objects: path, walls, frigid, river, rainbow, wall, all
Action: west
Score: 5
---
Obs: Desc: Forest This is a dimly lit forest, with large trees all around. Inv: You are emptyhanded. Feedback:
      Forest This is a dimly lit forest, with large trees all around.
Template probs: south: 0.494, open OBJ: 0.241, east: 0.105, west: 0.053, north: 0.038, go around OBJ: 0.018,
     northwest: 0.015, blow out OBJ: 0.014, throw OBJ at OBJ: 0.007, down: 0.005
Object probs: window: 0.699, small: 0.301
Interactive objects: forest, large, all, trees
Action: south
Score: 5
---
Obs: Desc: Forest This is a dimly lit forest, with large trees all around. Inv: You are emptyhanded. Feedback:
      Stormtossed trees block your way.
Template probs: east: 0.722, north: 0.067, west: 0.061, open OBJ: 0.061, south: 0.048, blow out OBJ: 0.011,
     northwest: 0.01, go around OBJ: 0.007, throw OBJ at OBJ: 0.005
Object probs: window: 0.7, small: 0.3
Interactive objects: storm, forest, trees, large, all
Action: east
Score: 5
---
Obs: Desc: Forest This is a dimly lit forest, with large trees all around. Inv: You are emptyhanded. Feedback:
      The rank undergrowth prevents eastward movement.
Template probs: south: 0.434, open OBJ: 0.339, east: 0.076, west: 0.065, north: 0.024, go around OBJ: 0.02,
     northwest: 0.012, blow out OBJ: 0.011, throw OBJ at OBJ: 0.009
Object probs: window: 0.701, small: 0.299
```

```
Interactive objects: forest, large, all, trees
Action: jump
Score: 5
---
Obs: Desc: Forest This is a dimly lit forest, with large trees all around. Inv: You are emptyhanded. Feedback:
    Wheeeeeeeeeee!!!!! You hear in the distance the chirping of a song bird.
Template probs: south: 0.438, open OBJ: 0.292, east: 0.098, west: 0.075, north: 0.03, go around OBJ: 0.019,
    northwest: 0.013, blow out OBJ: 0.012, throw OBJ at OBJ: 0.01
Object probs: window: 0.7, small: 0.299
Interactive objects: forest, large, all, trees
Action: south
Score: 5
---
Obs: Desc: Forest This is a dimly lit forest, with large trees all around. Inv: You are emptyhanded. Feedback:
    Stormtossed trees block your way.
Template probs: east: 0.694, west: 0.081, north: 0.07, open OBJ: 0.067, south: 0.043, northwest: 0.012, blow
    out OBJ: 0.011, go around OBJ: 0.007, throw OBJ at OBJ: 0.006
Object probs: window: 0.7, small: 0.3
Interactive objects: storm, forest, trees, large, all
Action: north
Score: 5
---
Obs: Desc: Clearing You are in a small clearing in a well marked forest path that extends to the east and west
    . Inv: You are emptyhanded. Feedback: Clearing
Template probs: west: 0.437, east: 0.311, open OBJ: 0.171, north: 0.026, throw OBJ at OBJ: 0.015, south:
    0.013, blow out OBJ: 0.011, go around OBJ: 0.006, northwest: 0.006
Object probs: window: 0.699, small: 0.301
Interactive objects: forest, path, all
Action: throw path at forest
Score: 5
---
Obs: Desc: Clearing You are in a small clearing in a well marked forest path that extends to the east and west
    . Inv: You are emptyhanded. Feedback: You dont have the way.
Template probs: open OBJ: 0.377, west: 0.341, east: 0.149, south: 0.057, north: 0.022, throw OBJ at OBJ:
    0.018, blow out OBJ: 0.011, go around OBJ: 0.011, northwest: 0.008
Object probs: window: 0.7, small: 0.299
Interactive objects: forest, path, all
Action: west
Score: 5
---
Obs: Desc: Behind House You are behind the white house. A path leads into the forest to the east. In one
    corner of the house there is a small window which is slightly ajar. Inv: You are emptyhanded. Feedback:
    Behind House You are behind the white house. A path leads into the forest to the east. In one corner of
    the house there is a small window which is slightly ajar.
Template probs: open OBJ: 0.953, west: 0.036, south: 0.006
Object probs: small: 0.898, window: 0.102
Interactive objects: path, window, small, white, house, all
Action: open house
Score: 5
---
Obs: Desc: Behind House You are behind the white house. A path leads into the forest to the east. In one
    corner of the house there is a small window which is slightly ajar. Inv: You are emptyhanded. Feedback:
    The window is closed.
Template probs: open OBJ: 0.943, west: 0.047
Object probs: small: 0.88, window: 0.12
Interactive objects: path, window, small, white, house, all
Action: open path
Score: 5
---
Obs: Desc: Behind House You are behind the white house. A path leads into the forest to the east. In one
    corner of the house there is a small window which is slightly ajar. Inv: You are emptyhanded. Feedback:
    You must tell me how to do that to a way.
Template probs: open OBJ: 0.899, east: 0.048, west: 0.044, south: 0.004
Object probs: window: 0.58, small: 0.417, kitche: 0.002
Interactive objects: path, window, small, white, house, all
Action: open window
Score: 5
---
Obs: Desc: Behind House You are behind the white house. A path leads into the forest to the east. In one
    corner of the house there is a small window which is open. Inv: You are emptyhanded. Feedback: With
    great effort, you open the window far enough to allow entry.
Template probs: west: 0.997
Object probs: small: 0.951, window: 0.049
Interactive objects: path, window, small, white, house, all
Action: west
Score: 5
---
Obs: Desc: Kitchen You are in the kitchen of the white house. A table seems to have been used recently for the
    preparation of food. A passage leads to the west and a dark staircase can be seen leading upward. A
    dark chimney leads down and to the east is a small window which is open. On the table is an elongated
    brown sack, smelling of hot peppers. A bottle is sitting on the table. The glass bottle contains A
    quantity of water Inv: You are emptyhanded. Feedback: Kitchen You are in the kitchen of the white house.
    A table seems to have been used recently for the preparation of food. A passage leads to the west and a
    dark staircase can be seen leading upward. A dark chimney leads down and to the east is a small window
    which is open. On the table is an elongated brown sack, smelling of hot peppers. A bottle is sitting on
    the table. The glass bottle contains A quantity of water
Template probs: take OBJ: 0.47, west: 0.351, turn on OBJ: 0.061, blow out OBJ: 0.052, down: 0.023, open OBJ:
    0.012, east: 0.008, close OBJ: 0.006, throw OBJ at OBJ: 0.006
Object probs: window: 0.683, small: 0.317
Interactive objects: kitchen, window, passage, staircase, glass, sack, water, small, table, chimney, bottle,
    quantity, brown, all
Action: west
```

```
Score: 15
---
Obs: Desc: Living Room You are in the living room. There is a doorway to the east, a wooden door with strange
    gothic lettering to the west, which appears to be nailed shut, a trophy case, and a large oriental rug
    in the center of the room. Above the trophy case hangs an elvish sword of great antiquity. A
    batterypowered brass lantern is on the trophy case. Inv: You are emptyhanded. Feedback: Living Room You
    are in the living room. There is a doorway to the east, a wooden door with strange gothic lettering to
    the west, which appears to be nailed shut, a trophy case, and a large oriental rug in the center of the
    room. Above the trophy case hangs an elvish sword of great antiquity. A batterypowered brass lantern is
    on the trophy case.
Template probs: take OBJ: 0.454, west: 0.241, turn on OBJ: 0.107, open OBJ: 0.053, down: 0.05, blow out OBJ:
    0.034, close OBJ: 0.016, throw OBJ at OBJ: 0.011, east: 0.009, north: 0.005
Object probs: window: 0.685, small: 0.315
Interactive objects: elvish, strange, trophy, brass, wooden, rug, oriental, lettering, antiquity, sword,
    gothic, west, door, large, lantern, case, all
Action: take brass
Score: 15
---
Obs: Desc: Living Room You are in the living room. There is a doorway to the east, a wooden door with strange
    gothic lettering to the west, which appears to be nailed shut, a trophy case, and a large oriental rug
    in the center of the room. Above the trophy case hangs an elvish sword of great antiquity. Inv: You are
    carrying A brass lantern Feedback: Taken.
Template probs: open OBJ: 0.48, push OBJ: 0.138, west: 0.088, take OBJ: 0.067, turn on OBJ: 0.059, throw OBJ
    at OBJ: 0.031, put down OBJ: 0.023, put OBJ in OBJ: 0.018, put OBJ on OBJ: 0.016, down: 0.015
Object probs: window: 0.685, small: 0.315
Interactive objects: elvish, strange, trophy, brass, wooden, rug, oriental, lettering, antiquity, sword,
    gothic, west, door, large, lantern, case, all
Action: turn on brass
Score: 15
---
Obs: Desc: Living Room You are in the living room. There is a doorway to the east, a wooden door with strange
    gothic lettering to the west, which appears to be nailed shut, a trophy case, and a large oriental rug
    in the center of the room. Above the trophy case hangs an elvish sword of great antiquity. Inv: You are
    carrying A brass lantern (providing light) Feedback: The brass lantern is now on.
Template probs: open OBJ: 0.438, push OBJ: 0.141, west: 0.088, take OBJ: 0.081, turn on OBJ: 0.061, throw OBJ
    at OBJ: 0.036, put down OBJ: 0.023, put OBJ in OBJ: 0.019, down: 0.018, put OBJ on OBJ: 0.018
Object probs: window: 0.685, small: 0.315
Interactive objects: brass, strange, trophy, elvish, rug, oriental, light, lettering, antiquity, sword, gothic
    , west, door, large, wooden, lantern, case, all
Action: take trophy
Score: 15
---
Obs: Desc: Living Room You are in the living room. There is a doorway to the east, a wooden door with strange
    gothic lettering to the west, which appears to be nailed shut, a trophy case, and a large oriental rug
    in the center of the room. Above the trophy case hangs an elvish sword of great antiquity. Inv: You are
    carrying A brass lantern (providing light) Feedback: The trophy case is securely fastened to the wall.
Template probs: open OBJ: 0.437, push OBJ: 0.141, west: 0.087, take OBJ: 0.082, turn on OBJ: 0.061, throw OBJ
    at OBJ: 0.035, put down OBJ: 0.024, down: 0.019, put OBJ on OBJ: 0.018, put OBJ in OBJ: 0.018
Object probs: window: 0.684, small: 0.315
Interactive objects: strange, trophy, case, antiquity, lantern, lettering, sword, gothic, wooden, wall, elvish
    , brass, west, rug, oriental, door, large, light, all
Action: push case
Score: 15
---
Obs: Desc: Living Room You are in the living room. There is a doorway to the east, a wooden door with strange
    gothic lettering to the west, which appears to be nailed shut, a trophy case, and a large oriental rug
    in the center of the room. Above the trophy case hangs an elvish sword of great antiquity. Inv: You are
    carrying A brass lantern (providing light) Feedback: Pushing the trophy case has no effect.
Template probs: open OBJ: 0.407, push OBJ: 0.145, take OBJ: 0.09, west: 0.085, turn on OBJ: 0.062, throw OBJ
    at OBJ: 0.037, put down OBJ: 0.024, down: 0.021, put OBJ on OBJ: 0.019, put OBJ in OBJ: 0.019
Object probs: window: 0.686, small: 0.314
Interactive objects: elvish, strange, trophy, brass, wooden, rug, oriental, case, lettering, antiquity, sword,
    gothic, west, door, large, lantern, light, all
Action: push lanter
Score: 15
---
Obs: Desc: Living Room You are in the living room. There is a doorway to the east, a wooden door with strange
    gothic lettering to the west, which appears to be nailed shut, a trophy case, and a large oriental rug
    in the center of the room. Above the trophy case hangs an elvish sword of great antiquity. Inv: You are
    carrying A brass lantern (providing light) Feedback: Pushing the brass lantern isnt notably helpful.
Template probs: open OBJ: 0.405, push OBJ: 0.144, west: 0.093, take OBJ: 0.089, turn on OBJ: 0.062, throw OBJ
    at OBJ: 0.038, put down OBJ: 0.025, put OBJ on OBJ: 0.02, down: 0.019, put OBJ in OBJ: 0.019
Object probs: window: 0.685, small: 0.315
Interactive objects: brass, strange, trophy, elvish, rug, oriental, light, lettering, antiquity, sword, gothic
    , west, door, large, wooden, lantern, case, all
Action: open wooden
Score: 15
---
Obs: Desc: Living Room You are in the living room. There is a doorway to the east, a wooden door with strange
    gothic lettering to the west, which appears to be nailed shut, a trophy case, and a large oriental rug
    in the center of the room. Above the trophy case hangs an elvish sword of great antiquity. Inv: You are
    carrying A brass lantern (providing light) Feedback: The door cannot be opened.
Template probs: open OBJ: 0.438, push OBJ: 0.142, take OBJ: 0.081, west: 0.08, turn on OBJ: 0.062, throw OBJ
    at OBJ: 0.035, put down OBJ: 0.024, down: 0.02, put OBJ in OBJ: 0.018, put OBJ on OBJ: 0.018
Object probs: window: 0.685, small: 0.315
Interactive objects: elvish, strange, trophy, brass, wooden, rug, oriental, case, lettering, antiquity, sword,
    gothic, west, door, large, lantern, light, all
Action: open brass
Score: 15
---
Obs: Desc: Living Room You are in the living room. There is a doorway to the east, a wooden door with strange
    gothic lettering to the west, which appears to be nailed shut, a trophy case, and a large oriental rug
```

```
           in the center of the room. Above the trophy case hangs an elvish sword of great antiquity. Inv: You are
           carrying A brass lantern (providing light) Feedback: You must tell me how to do that to a brass lantern.
Template probs: open OBJ: 0.503, push OBJ: 0.129, west: 0.1, take OBJ: 0.053, turn on OBJ: 0.051, throw OBJ at
           OBJ: 0.041, put down OBJ: 0.029, put OBJ in OBJ: 0.021, put OBJ on OBJ: 0.02, down: 0.009
Object probs: window: 0.677, small: 0.323
Interactive objects: brass, strange, trophy, elvish, rug, oriental, light, lettering, antiquity, sword, gothic
           , west, door, large, wooden, lantern, case, all
Action: push rug
Score: 15
---
Obs: Desc: Living Room You are in the living room. There is a doorway to the east, a wooden door with strange
           gothic lettering to the west, which appears to be nailed shut, a trophy case, and a closed trap door at
           your feet. Above the trophy case hangs an elvish sword of great antiquity. Inv: You are carrying A brass
           lantern (providing light) Feedback: With a great effort, the rug is moved to one side of the room,
           revealing the dusty cover of a closed trap door.
Template probs: open OBJ: 0.582, push OBJ: 0.082, down: 0.074, take OBJ: 0.065, put down OBJ: 0.025, turn on
           OBJ: 0.025, west: 0.023, close OBJ: 0.021, throw OBJ at OBJ: 0.019, put OBJ in OBJ: 0.016
Object probs: window: 0.674, small: 0.326
Interactive objects: dusty, strange, trophy, cover, case, antiquity, lantern, lettering, sword, gothic, wooden
           , elvish, brass, west, rug, trap, door, light, all
Action: west
Score: 15
---
Obs: Desc: Living Room You are in the living room. There is a doorway to the east, a wooden door with strange
           gothic lettering to the west, which appears to be nailed shut, a trophy case, and a closed trap door at
           your feet. Above the trophy case hangs an elvish sword of great antiquity. Inv: You are carrying A brass
           lantern (providing light) Feedback: The door is nailed shut.
Template probs: open OBJ: 0.679, push OBJ: 0.073, down: 0.057, take OBJ: 0.046, put down OBJ: 0.022, turn on
           OBJ: 0.019, west: 0.017, close OBJ: 0.015, throw OBJ at OBJ: 0.015, put OBJ in OBJ: 0.015
Object probs: window: 0.672, small: 0.328
Interactive objects: elvish, strange, trophy, brass, case, lettering, antiquity, sword, gothic, west, door,
           trap, wooden, lantern, light, all
Action: down
Score: 15
---
Obs: Desc: Living Room You are in the living room. There is a doorway to the east, a wooden door with strange
           gothic lettering to the west, which appears to be nailed shut, a trophy case, and a closed trap door at
           your feet. Above the trophy case hangs an elvish sword of great antiquity. Inv: You are carrying A brass
           lantern (providing light) Feedback: The trap door is closed.
Template probs: open OBJ: 0.63, down: 0.087, take OBJ: 0.06, put down OBJ: 0.045, push OBJ: 0.04, west: 0.027,
           turn on OBJ: 0.02, close OBJ: 0.018, throw OBJ at OBJ: 0.018, put OBJ in OBJ: 0.015
Object probs: window: 0.657, small: 0.342
Interactive objects: elvish, strange, trophy, brass, case, lettering, antiquity, sword, gothic, west, trap,
           door, wooden, lantern, light, all
Action: open sword
Score: 15
---
Obs: Desc: Living Room You are in the living room. There is a doorway to the east, a wooden door with strange
           gothic lettering to the west, which appears to be nailed shut, a trophy case, and a closed trap door at
           your feet. Above the trophy case hangs an elvish sword of great antiquity. Inv: You are carrying A brass
           lantern (providing light) Feedback: You must tell me how to do that to a sword.
Template probs: open OBJ: 0.625, down: 0.075, take OBJ: 0.053, push OBJ: 0.05, put down OBJ: 0.049, west:
           0.028, turn on OBJ: 0.021, throw OBJ at OBJ: 0.02, close OBJ: 0.019, put OBJ in OBJ: 0.018
Object probs: window: 0.659, small: 0.34
Interactive objects: elvish, strange, trophy, brass, case, lettering, antiquity, sword, gothic, west, door,
           trap, wooden, lantern, light, all
Action: open case
Score: 15
---
Obs: Desc: Living Room You are in the living room. There is a doorway to the east, a wooden door with strange
           gothic lettering to the west, which appears to be nailed shut, a trophy case, and a closed trap door at
           your feet. Above the trophy case hangs an elvish sword of great antiquity. Inv: You are carrying A brass
           lantern (providing light) Feedback: Opened.
Template probs: open OBJ: 0.61, down: 0.081, take OBJ: 0.056, push OBJ: 0.053, put down OBJ: 0.049, west:
           0.029, turn on OBJ: 0.023, throw OBJ at OBJ: 0.02, put OBJ in OBJ: 0.019, close OBJ: 0.019
Object probs: window: 0.659, small: 0.341
Interactive objects: elvish, strange, trophy, brass, case, lettering, antiquity, sword, gothic, west, door,
           trap, wooden, lantern, light, all
Action: open door
Score: 15
---
Obs: Desc: Living Room You are in the living room. There is a doorway to the east, a wooden door with strange
           gothic lettering to the west, which appears to be nailed shut, a trophy case, and a closed trap door at
           your feet. Above the trophy case hangs an elvish sword of great antiquity. Inv: You are carrying A brass
           lantern (providing light) Feedback: Which door do you mean, the wooden door or the trap door?
Template probs: open OBJ: 0.552, down: 0.085, take OBJ: 0.062, put down OBJ: 0.06, push OBJ: 0.055, west:
           0.04, turn on OBJ: 0.026, throw OBJ at OBJ: 0.026, put OBJ in OBJ: 0.024, close OBJ: 0.021
Object probs: window: 0.66, small: 0.33
Interactive objects: elvish, strange, trophy, brass, case, lettering, antiquity, sword, gothic, west, trap,
           door, wooden, lantern, light, all
Action: open all
Score: 15
---
Obs: Desc: Living Room You are in the living room. There is a doorway to the east, a wooden door with strange
           gothic lettering to the west, which appears to be nailed shut, a trophy case, and a closed trap door at
           your feet. Above the trophy case hangs an elvish sword of great antiquity. Inv: You are carrying A brass
           lantern (providing light) Feedback: You cant use multiple direct objects with "open".
Template probs: open OBJ: 0.529, down: 0.086, put down OBJ: 0.065, take OBJ: 0.063, push OBJ: 0.059, west:
           0.044, throw OBJ at OBJ: 0.028, turn on OBJ: 0.028, put OBJ in OBJ: 0.026, close OBJ: 0.021
Object probs: window: 0.661, small: 0.339
Interactive objects: elvish, strange, trophy, brass, case, lettering, antiquity, sword, gothic, west, door,
           trap, wooden, lantern, light, all
```

```
Action: down
Score: 15
---
Obs: Desc: Living Room You are in the living room. There is a doorway to the east, a wooden door with strange
     gothic lettering to the west, which appears to be nailed shut, a trophy case, and a closed trap door at
     your feet. Above the trophy case hangs an elvish sword of great antiquity. Inv: You are carrying A brass
     lantern (providing light) Feedback: The trap door is closed.
Template probs: open OBJ: 0.569, down: 0.089, take OBJ: 0.063, put down OBJ: 0.059, push OBJ: 0.048, west:
     0.036, turn on OBJ: 0.024, throw OBJ at OBJ: 0.024, put OBJ in OBJ: 0.021, close OBJ: 0.021
Object probs: window: 0.658, small: 0.341
Interactive objects: elvish, strange, trophy, brass, case, lettering, antiquity, sword, gothic, west, trap,
     door, wooden, lantern, light, all
Action: shake gothic
Score: 15
---
Obs: Desc: Living Room You are in the living room. There is a doorway to the east, a wooden door with strange
     gothic lettering to the west, which appears to be nailed shut, a trophy case, and a closed trap door at
     your feet. Above the trophy case hangs an elvish sword of great antiquity. Inv: You are carrying A brass
     lantern (providing light) Feedback: You dont have the wooden door.
Template probs: open OBJ: 0.62, push OBJ: 0.085, take OBJ: 0.055, west: 0.031, put down OBJ:
     0.026, turn on OBJ: 0.025, throw OBJ at OBJ: 0.02, put OBJ in OBJ: 0.019, close OBJ: 0.014
Object probs: window: 0.668, small: 0.332
Interactive objects: elvish, strange, trophy, brass, case, lettering, antiquity, sword, gothic, west, door,
     trap, wooden, lantern, light, all
Action: open west
Score: 15
---
Obs: Desc: Living Room You are in the living room. There is a doorway to the east, a wooden door with strange
     gothic lettering to the west, which appears to be nailed shut, a trophy case, and a closed trap door at
     your feet. Above the trophy case hangs an elvish sword of great antiquity. Inv: You are carrying A brass
     lantern (providing light) Feedback: The door cannot be opened.
Template probs: open OBJ: 0.648, push OBJ: 0.089, down: 0.053, take OBJ: 0.042, west: 0.025, turn on OBJ:
     0.025, put down OBJ: 0.024, throw OBJ at OBJ: 0.021, put OBJ in OBJ: 0.02, close OBJ: 0.013
Object probs: window: 0.665, small: 0.335
Interactive objects: elvish, strange, trophy, brass, case, lettering, antiquity, sword, gothic, west, door,
     trap, wooden, lantern, light, all
Action: open wooden
Score: 15
---
Obs: Desc: Living Room You are in the living room. There is a doorway to the east, a wooden door with strange
     gothic lettering to the west, which appears to be nailed shut, a trophy case, and a closed trap door at
     your feet. Above the trophy case hangs an elvish sword of great antiquity. Inv: You are carrying A brass
     lantern (providing light) Feedback: The door cannot be opened.
Template probs: open OBJ: 0.637, push OBJ: 0.081, take OBJ: 0.05, put down OBJ: 0.024, west:
     0.024, turn on OBJ: 0.024, close OBJ: 0.019, throw OBJ at OBJ: 0.017, put OBJ in OBJ: 0.016
Object probs: window: 0.671, small: 0.329
Interactive objects: elvish, strange, trophy, brass, case, lettering, antiquity, sword, gothic, west, door,
     trap, wooden, lantern, light, all
Action: push sword
Score: 15
---
Obs: Desc: Living Room You are in the living room. There is a doorway to the east, a wooden door with strange
     gothic lettering to the west, which appears to be nailed shut, a trophy case, and a closed trap door at
     your feet. Above the trophy case hangs an elvish sword of great antiquity. Inv: You are carrying A brass
     lantern (providing light) Feedback: Pushing the sword doesnt seem to work.
Template probs: open OBJ: 0.591, push OBJ: 0.085, down: 0.073, take OBJ: 0.061, turn on OBJ: 0.025, put down
     OBJ: 0.024, west: 0.024, close OBJ: 0.02, throw OBJ at OBJ: 0.019, put OBJ in OBJ: 0.017
Object probs: window: 0.674, small: 0.326
Interactive objects: elvish, strange, trophy, brass, case, lettering, antiquity, sword, gothic, west, door,
     trap, wooden, lantern, light, all
Action: open strang
Score: 15
---
Obs: Desc: Living Room You are in the living room. There is a doorway to the east, a wooden door with strange
     gothic lettering to the west, which appears to be nailed shut, a trophy case, and a closed trap door at
     your feet. Above the trophy case hangs an elvish sword of great antiquity. Inv: You are carrying A brass
     lantern (providing light) Feedback: The door cannot be opened.
Template probs: open OBJ: 0.614, push OBJ: 0.085, down: 0.065, take OBJ: 0.059, west: 0.024, turn on OBJ:
     0.024, put down OBJ: 0.021, close OBJ: 0.017, throw OBJ at OBJ: 0.017, put OBJ in OBJ: 0.016
Object probs: window: 0.676, small: 0.323
Interactive objects: elvish, strange, trophy, brass, case, lettering, antiquity, sword, gothic, west, door,
     trap, wooden, lantern, light, all
Action: down
Score: 15
---
Obs: Desc: Living Room You are in the living room. There is a doorway to the east, a wooden door with strange
     gothic lettering to the west, which appears to be nailed shut, a trophy case, and a closed trap door at
     your feet. Above the trophy case hangs an elvish sword of great antiquity. Inv: You are carrying A brass
     lantern (providing light) Feedback: The trap door is closed.
Template probs: open OBJ: 0.665, down: 0.068, push OBJ: 0.052, take OBJ: 0.044, put down OBJ: 0.036, west:
     0.026, turn on OBJ: 0.02, throw OBJ at OBJ: 0.018, put OBJ in OBJ: 0.017, close OBJ: 0.015
Object probs: window: 0.661, small: 0.339
Interactive objects: elvish, strange, trophy, brass, case, lettering, antiquity, sword, gothic, west, trap,
     door, wooden, lantern, light, all
Action: open trap
Score: 15
---
Obs: Desc: Living Room You are in the living room. There is a doorway to the east, a wooden door with strange
     gothic lettering to the west, which appears to be nailed shut, a trophy case, and a rug lying beside an
     open trap door. Above the trophy case hangs an elvish sword of great antiquity. Inv: You are carrying A
     brass lantern (providing light) Feedback: The door reluctantly opens to reveal a rickety staircase
     descending into darkness.
```

```
Template probs: down: 0.876, open OBJ: 0.046, take OBJ: 0.018, east: 0.013, close OBJ: 0.009, put down OBJ:
    0.007, north: 0.006
Object probs: window: 0.604, small: 0.387, kitche: 0.006, trap: 0.003
Interactive objects: elvish, strange, trophy, brass, rug, staircase, case, lettering, antiquity, sword, gothic
    , west, door, trap, wooden, lantern, light, all
Action: down
Score: 15
---
Obs: Desc: It is pitch black. You are likely to be eaten by a grue. Inv: You are carrying A brass lantern
    Feedback: You have moved into a dark place. It is pitch black. You are likely to be eaten by a grue.
Template probs: take OBJ: 0.141, turn on OBJ: 0.14, open OBJ: 0.122, down: 0.117, close OBJ: 0.073, put down
    OBJ: 0.069, west: 0.065, push OBJ: 0.056, throw OBJ at OBJ: 0.054, south: 0.02
Object probs: lanter: 0.076, letter: 0.071, window: 0.061, door: 0.059, small: 0.052, gothic: 0.049, sword:
    0.049, rug: 0.037, trophy: 0.035, west: 0.031
Interactive objects: brass, dark, lantern, black, all, grue
Action: down
Score: 40
---
Obs: Desc: It is pitch black. You are likely to be eaten by a grue. Inv: You are carrying A brass lantern
    Feedback: The chasm probably leads straight to the infernal regions.
Template probs: turn on OBJ: 0.167, take OBJ: 0.132, open OBJ: 0.125, down: 0.124, close OBJ: 0.076, push OBJ:
    0.067, put down OBJ: 0.066, throw OBJ at OBJ: 0.056, west: 0.05, south: 0.024
Object probs: lanter: 0.077, letter: 0.071, window: 0.06, door: 0.059, small: 0.051, gothic: 0.05, sword:
    0.049, rug: 0.037, trophy: 0.035, west: 0.031
Interactive objects: brass, chasm, lantern, black, all, grue
Action: turn on lanter
Score: 40
---
Obs: Desc: Cellar You are in a dark and damp cellar with a narrow passageway leading north, and a crawlway to
    the south. On the west is the bottom of a steep metal ramp which is unclimbable. There is a
    suspiciouslooking individual, holding a large bag, leaning against one wall. He is armed with a deadly
    stiletto. Inv: You are carrying A brass lantern (providing light) Feedback: You try to ascend the ramp,
    but it is impossible, and you slide back down.
Template probs: down: 0.246, open OBJ: 0.209, take OBJ: 0.122, close OBJ: 0.1, push OBJ: 0.082, turn on OBJ:
    0.068, put down OBJ: 0.033, south: 0.029, west: 0.019, throw OBJ at OBJ: 0.016
Object probs: west: 0.08, small: 0.08, window: 0.073, door: 0.067, rug: 0.057, letter: 0.047, wooden: 0.045,
    gothic: 0.041, strang: 0.035, lanter: 0.034
Interactive objects: light, ramp, large, brass, stiletto, crawlway, dark, narrow, lantern, bag, suspicious,
    metal, all, wall, passageway
Action: take dark
Score: 40
---
Obs: Desc: Cellar You are in a dark and damp cellar with a narrow passageway leading north, and a crawlway to
    the south. On the west is the bottom of a steep metal ramp which is unclimbable. There is a
    suspiciouslooking individual, holding a large bag, leaning against one wall. He is armed with a deadly
    stiletto. Inv: You are carrying A brass lantern (providing light) Feedback: You cant be serious.
Template probs: down: 0.289, open OBJ: 0.201, take OBJ: 0.103, close OBJ: 0.091, turn on OBJ: 0.066, push OBJ:
    0.065, put down OBJ: 0.043, throw OBJ at OBJ: 0.022, south: 0.021, west: 0.018
Object probs: west: 0.081, small: 0.076, window: 0.069, door: 0.068, rug: 0.057, letter: 0.048, wooden: 0.045,
    gothic: 0.042, strang: 0.036, lanter: 0.035
Interactive objects: light, ramp, large, brass, stiletto, crawlway, dark, narrow, lantern, bag, suspicious,
    metal, all, wall, passageway
Action: put out passag
Score: 40
---
Obs: Desc: Cellar You are in a dark and damp cellar with a narrow passageway leading north, and a crawlway to
    the south. On the west is the bottom of a steep metal ramp which is unclimbable. There is a
    suspiciouslooking individual, holding a large bag, leaning against one wall. He is armed with a deadly
    stiletto. The thief, finding nothing of value, left disgusted. Inv: You are carrying A brass lantern (
    providing light) The thief, finding nothing of value, left disgusted. Feedback: Pushing the chute isnt
    notably helpful. Someone carrying a large bag is casually leaning against one of the walls here. He does
    not speak, but it is clear from his aspect that the bag will be taken only over his dead body.
Template probs: open OBJ: 0.211, take OBJ: 0.205, push OBJ: 0.188, south: 0.132, west: 0.105, close OBJ:
    0.057, turn on OBJ: 0.026, go around OBJ: 0.02, east: 0.017, down: 0.016
Object probs: small: 0.112, window: 0.103, west: 0.075, door: 0.063, rug: 0.054, wooden: 0.044, letter: 0.04,
    gothic: 0.037, strang: 0.032, kitche: 0.031
Interactive objects: light, ramp, large, brass, stiletto, crawlway, walls, dark, narrow, thief, lantern, chute
    , bag, suspicious, metal, all, wall, passageway
Action: west
Score: 40
---
Obs: Desc: Cellar You are in a dark and damp cellar with a narrow passageway leading north, and a crawlway to
    the south. On the west is the bottom of a steep metal ramp which is unclimbable. There is a
    suspiciouslooking individual, holding a large bag, leaning against one wall. He is armed with a deadly
    stiletto. A quick thrust pinks your left arm, and blood starts to trickle down. Inv: You are carrying A
    brass lantern (providing light) The thief, finding nothing of value, left disgusted. Feedback: You dont
    have the way.
Template probs: take OBJ: 0.246, open OBJ: 0.211, push OBJ: 0.19, south: 0.091, west: 0.082, close OBJ: 0.059,
    turn on OBJ: 0.041, down: 0.026, go around OBJ: 0.017, east: 0.012
Object probs: small: 0.097, window: 0.087, west: 0.078, door: 0.065, rug: 0.056, wooden: 0.045, letter: 0.043,
    gothic: 0.038, strang: 0.034, lanter: 0.032
Interactive objects: light, ramp, large, brass, stiletto, crawlway, dark, narrow, lantern, bag, suspicious,
    metal, all, wall, passageway
Action: south
Score: 40
---
Obs: Desc: East of Chasm You are on the east edge of a chasm, the bottom of which cannot be seen. A narrow
    passage goes north, and the path you are on continues to the east. Inv: You are carrying A brass lantern
    (providing light) Feedback: The brass lantern is now on. East of Chasm You are on the east edge of a
    chasm, the bottom of which cannot be seen. A narrow passage goes north, and the path you are on
    continues to the east.
```

```
Template probs: put down OBJ: 0.132, take OBJ: 0.116, east: 0.1, open OBJ: 0.093, close OBJ: 0.09, turn on OBJ
    : 0.07, throw OBJ at OBJ: 0.061, west: 0.059, push OBJ: 0.057, down: 0.042
Object probs: window: 0.074, small: 0.057, lanter: 0.046, sword: 0.041, gothic: 0.034, wooden: 0.031, letter:
    0.03, bottle: 0.03, door: 0.028, large: 0.028
Interactive objects: light, brass, chasm, path, narrow, lantern, passage, all
Action: east
Score: 40
---
Obs: Desc: Gallery This is an art gallery. Most of the paintings have been stolen by vandals with exceptional
    taste. The vandals left through either the north or west exits. Fortunately, there is still one chance
    for you to be a vandal, for on the far wall is a painting of unparalleled beauty. Inv: You are
    emptyhanded. Feedback: Gallery This is an art gallery. Most of the paintings have been stolen by vandals
    with exceptional taste. The vandals left through either the north or west exits. Fortunately, there is
    still one chance for you to be a vandal, for on the far wall is a painting of unparalleled beauty.
Template probs: open OBJ: 0.42, west: 0.377, take OBJ: 0.063, blow out OBJ: 0.039, east: 0.026, push OBJ:
    0.02, down: 0.016, go around OBJ: 0.01, north: 0.008, turn on OBJ: 0.006
Object probs: window: 0.701, small: 0.299
Interactive objects: art, painting, wall, all, paintings
Action: open painti
Score: 40
---
Obs: Desc: Gallery This is an art gallery. Most of the paintings have been stolen by vandals with exceptional
    taste. The vandals left through either the north or west exits. Fortunately, there is still one chance
    for you to be a vandal, for on the far wall is a painting of unparalleled beauty. Inv: You are
    emptyhanded. Feedback: You must tell me how to do that to a painting.
Template probs: open OBJ: 0.41, west: 0.33, take OBJ: 0.106, push OBJ: 0.033, blow out OBJ: 0.032, down:
    0.023, turn on OBJ: 0.015, east: 0.014, go around OBJ: 0.01, south: 0.006
Object probs: window: 0.701, small: 0.299
Interactive objects: art, painting, wall, all, paintings
Action: take painti
Score: 40
---
Obs: Desc: Gallery This is an art gallery. Most of the paintings have been stolen by vandals with exceptional
    taste. The vandals left through either the north or west exits. Inv: You are carrying A painting
    Feedback: Taken.
Template probs: open OBJ: 0.492, push OBJ: 0.296, take OBJ: 0.036, west: 0.033, turn on OBJ: 0.032, throw OBJ
    at OBJ: 0.021, down: 0.015, close OBJ: 0.013, put down OBJ: 0.011, put OBJ in OBJ: 0.01
Object probs: window: 0.698, small: 0.302
Interactive objects: art, painting, all, paintings
Action: open art
Score: 44
---
```

## D.1 PENTARI TRANSCRIPT

```
TmplPred: examine OBJ 0.157, put OBJ down 0.129, take OBJ 0.072, sit on top of OBJ 0.067, listen to OBJ 0.063
ObjtPred: dirty 0.339, dagger 0.280, sword 0.144, chest 0.092, all 0.071
Act: put dagger down, Rew 0, Score 0, Done False, Value 0.339
Obs: A beautiful day in Bostwin! You are awakened by a ray of sunlight peeking through a window shade and rise
    refreshed after a restful night's sleep. You've given your Company a weekend pass to get some R&R after
    your last grueling mission. Wasting precious little time your entire company took off for the city
    proper to pack in as much drinking, carousing, brawling, wenching and other relaxing pursuits as
    possible in 48 hours. In the fields of battle your men are an elite unit of trained professionals. Off
    duty they're pretty much like every other bunch of soldiers you've ever served with. A weekend off and
    what to do? After the challenges of the past year when you were first promoted and assumed command of
    Charlie Company you've nearly forgotton what free time is and what you're supposed to do with it.
    Pentari An Interactive Prequel To First Light Copyright 1998 (C) Howard A. Sherman. All Rights Reserved.
    Type 'Info' For Information About This Game or 'Help' for Instructions. Release 6 / Serial number
    030206 / Inform v6.21 Library 6/10 Your Quarters Rank certainly does have its privileges. When you were
    a lowly lieutenant you had to share cramped quarters with a fellow lieutenant at your previous
    assignment. Commanding a company of your own affords you the luxury of private, spacious quarters just
    outside the barracks themselves. As a further abuse of your power your chest and bunk are not standard
    issue items from the Pentari millitary supply center. You can see a postcard here.
TmplPred: north 0.746, put OBJ down 0.127, take OBJ 0.126, examine OBJ 0.001, south 0.000
ObjtPred: postcard 0.817, sword 0.180, dagger 0.001, wall 0.001, elf 0.000
Act: put postcard down, Rew 0, Score 0, Done False, Value 0.634
Obs: The postcard is already here.
TmplPred: north 0.785, put OBJ down 0.108, take OBJ 0.105, examine OBJ 0.002, south 0.000
ObjtPred: postcard 0.620, sword 0.361, dagger 0.006, all 0.003, emerald 0.003
Act: north, Rew 0, Score 0, Done False, Value 0.633
Obs: Charlie Company Compound Your company compound consists of three large buildings bordering a large, open
    area where company formations precede the orders of the day. To the west the Company's supply building
    can be seen while directly ahead of you lies your company's barracks which seem quieter than a crypt.
    Off to the east you hear the distant shouts of another Company drilling. Poor souls.
TmplPred: north 0.915, put OBJ down 0.043, south 0.020, take OBJ 0.010, examine OBJ 0.005
ObjtPred: sword 0.982, postcard 0.014, dagger 0.002, all 0.001, enemy 0.000
Act: north, Rew 0, Score 0, Done False, Value 1.284
Obs: Charlie Company Barracks If the Army of Bostwin were to publish a new edition of the soldier's handbook,
    a picture of this barracks should be included as the example other companies should follow. The floors
    are polished to an almost blinding shine while each of the bunks lining both sides of this building are
    perfectly made. Every locker next to each bunk is closed and locked. Not a single scrap of paper, not
    one article of clothing, nay not even a dust particle looks out of place. The powerful smell of high
    quantities of pine cleaner nearly burns your nostrils. Damn, you run a good comapny. You can see a
    Pentarian Transporter here.
TmplPred: enter 0.974, get in OBJ 0.022, south 0.004, put OBJ down 0.001, get up 0.000
ObjtPred: transport 0.989, sword 0.011, legends 0.001, postcard 0.000, tapestrie 0.000
Act: enter, Rew 0, Score 0, Done False, Value 2.722
Obs: The Charlie Company Barracks (in the Pentarian Transporter)
TmplPred: city 0.974, get up 0.023, put OBJ down 0.003, take OBJ 0.000, south 0.000
ObjtPred: sword 0.996, postcard 0.002, transport 0.001, blade 0.000, dirty 0.000
```

```
Act: city, Rew 5, Score 5, Done False, Value 5.665
Obs: Your vision swims as your sight of the barracks blurs into a myriad of chaotic colors. You feel a moment
     of disorientation... As you begin to materialize your eyes grow wide as you realize you're not in front
     of the Bank of Bostwin, where the transporter usually takes you in the city. You're not even outdoors,
     you're indoors! What the hell is going on here...? Entrance Hall Your footsteps echo loudly as you walk
     through this room taking in the vast expanse of its size. The walls rise majestically upward and outward
     to form a huge room which, curiously, is empty. You can't help but wonder why such a large room is
     needed for the straight forward task of accepting guests and visitors. To the east a wide open arch
     leads to what must be a library while to the north you hardly help but notice a huge, semi circular
     marble staircase leading up to parts unknown. [Your score has just gone up by five points.]
TmplPred: east 0.783, put OBJ down 0.092, examine OBJ 0.076, north 0.026, take OBJ 0.016
ObjtPred: sword 0.630, dagger 0.169, emerald 0.097, all 0.051, east 0.033
Act: east, Rew 0, Score 5, Done False, Value 1.153
Obs: Library This is a vast chamber lined floor to ceiling with bookcases filled to capacity with books. You
     haven't seen such a collection of knowledge since your academy days. A cursory glance at some
     bookshelves reveal several titles on various topics of magic, history, philosophy, science and a hundred
     other topics. A few large tables with chairs in the center of the room inviting anyone to make use
     of these impressive facilities. The wizard Morden is here staring at you, the smallest of smiles on his
     face. You can also see a copy of The Legends and Lore of Pentari here.
TmplPred: covert 0.975, examine OBJ 0.008, put OBJ down 0.007, push OBJ to OBJ 0.005, take OBJ 0.002
ObjtPred: legends 0.700, sword 0.176, floor 0.062, dagger 0.029, all 0.010
Act: covert, Rew 0, Score 5, Done False, Value 2.394
Obs: Spots in all the colors of the rainbow dance in front of your eyes as you feel yourself displaced... You
     find yourself in the deep recesses of Vamvevmew's castle and instantly a wave of fear and depression hit
     you as hard as any enemy ever has. Your hand shoots right for your sword in its scabbard on reflex and
     you grip the hilt so hard your knuckles whiten. Castle You are at the center of an intersection which
     runs east to west as well as north to south. You can see The Dark Elf here. The Dark Elf walks around as
     if looking for something.
TmplPred: south 0.936, west 0.017, put OBJ down 0.015, north 0.010, east 0.008
ObjtPred: sword 0.643, dagger 0.127, dirty 0.084, enemy 0.052, seal 0.043
Act: south, Rew 0, Score 5, Done False, Value 5.257
Obs: Castle Entrance This large open area stands just inside the castle's gate and would be the place all
     visitors would first go before being allowed entrance to the castle proper. A guard station, long
     abandoned, is here. A corridor stretches into the castle itself to the north. You can see a Magical Seal
     here.
TmplPred: hit OBJ 0.947, put OBJ down 0.027, north 0.014, examine OBJ 0.005, south 0.005
ObjtPred: seal 0.999, ward 0.000, sword 0.000, postcard 0.000, dagger 0.000
Act: hit seal, Rew 10, Score 15, Done False, Value 10.024
Obs: Your fist smashes the ward, shattering it into a cloud of shimmering dust. [Your score has just gone up
     by ten points.]
TmplPred: north 0.802, put OBJ down 0.122, examine OBJ 0.075, take OBJ 0.001, south 0.000
ObjtPred: sword 0.850, postcard 0.061, blade 0.051, dirty 0.017, north 0.011
Act: north, Rew 0, Score 15, Done False, Value 0.450
Obs: Castle You can see The Dark Elf here. The Dark Elf walks around as if looking for something.
TmplPred: north 0.904, west 0.029, put OBJ down 0.024, south 0.019, examine OBJ 0.017
ObjtPred: sword 0.762, dagger 0.167, emerald 0.034, all 0.017, postcard 0.009
Act: north, Rew 0, Score 15, Done False, Value 1.362
Obs: Main Hall This once majestic room was where visitors would come to relax and meet with the formal lord of
     the castle in a somewhat informal atmosphere. Several large comfortable couches are scattered about,
     dusty and altogether squalid. Many large tapestries still hang on the walls but are horribly faded from
     age. Large open archways lead east and west while a huge fireplace dominates the center of the room
     against the northern wall.
TmplPred: west 0.938, north 0.045, put OBJ down 0.011, examine OBJ 0.003, south 0.001
ObjtPred: sword 0.735, tapestrie 0.110, east 0.052, all 0.033, wall 0.024
Act: west, Rew 0, Score 15, Done False, Value 2.757
Obs: Armory Many death dealing weapons of every type were stored here. Several tall racks probably held spears
     while shorter ones mounted against the wall stored various kinds of swords. Other wall mounts, also
     empty, give you no idea what sort of weapons may have been held by them. A large archway north is
     partially blocked by collapsed stones and rubble. You can see a jewel encrusted dagger here.
TmplPred: take OBJ 0.978, put OBJ down 0.009, examine OBJ 0.007, east 0.003, north 0.002
ObjtPred: all 0.887, dirty 0.034, dagger 0.031, other 0.028, emerald 0.017
Act: take all, Rew 5, Score 20, Done False, Value 5.732
Obs: jewel encrusted dagger: Taken. [Your score has just gone up by five points.]
TmplPred: east 0.924, examine OBJ 0.035, put OBJ down 0.029, north 0.007, take OBJ 0.003
ObjtPred: sword 0.357, other 0.278, dagger 0.111, all 0.108, postcard 0.036
Act: east, Rew 0, Score 20, Done False, Value 1.100
Obs: Main Hall You can see The Dark Elf here. The Dark Elf walks around as if looking for something.
TmplPred: north 0.928, put OBJ down 0.034, west 0.020, examine OBJ 0.011, south 0.004
ObjtPred: dagger 0.412, all 0.141, wall 0.128, sword 0.112, tapestrie 0.083
Act: north, Rew 0, Score 20, Done False, Value 2.549
Obs: Main Hall, by the Fireplace You are at the northern end of the main hall taking in the panorama of what
     was once a majestic area where perhaps treaties were signed, strategies laid and plans foiled. You can
     see a Large Fireplace here.
TmplPred: enter 0.991, put OBJ down 0.003, examine OBJ 0.002, south 0.002, sit on top of OBJ 0.001
ObjtPred: fireplace 0.637, all 0.122, dagger 0.103, sword 0.062, floor 0.020
Act: enter, Rew 0, Score 20, Done False, Value 5.182
Obs: The Main Hall, by the Fireplace (in the Large Fireplace) You can see a dirty scroll here.
TmplPred: take OBJ 0.914, get up 0.059, put OBJ down 0.019, examine OBJ 0.007, city 0.001
ObjtPred: dirty 0.938, all 0.051, dagger 0.008, sword 0.001, emerald 0.001
Act: take all, Rew 10, Score 30, Done False, Value 10.353
Obs: dirty scroll: Taken. [Your score has just gone up by ten points.]
TmplPred: get up 0.884, examine OBJ 0.046, fwoosh 0.036, put OBJ down 0.027, take OBJ 0.002
ObjtPred: dirty 0.535, sword 0.186, dagger 0.180, all 0.086, emerald 0.004
Act: get up, Rew 0, Score 30, Done False, Value 0.578
Obs: Main Hall, by the Fireplace You can see a Large Fireplace here.
TmplPred: south 0.938, enter 0.029, examine OBJ 0.014, put OBJ down 0.014, take OBJ 0.002
ObjtPred: dirty 0.305, sword 0.230, all 0.182, dagger 0.154, fireplace 0.099
Act: south, Rew 0, Score 30, Done False, Value 1.176
Obs: Main Hall
TmplPred: west 0.859, north 0.066, south 0.016, take OBJ 0.011, put OBJ down 0.009
ObjtPred: sword 0.435, dagger 0.253, dirty 0.168, all 0.103, east 0.009
Act: west, Rew 0, Score 30, Done False, Value 2.503
```

```
Obs: Armory
TmplPred: north 0.984, put OBJ down 0.008, examine OBJ 0.003, east 0.002, take OBJ 0.001
ObjtPred: dagger 0.387, dirty 0.283, all 0.200, wall 0.050, sword 0.043
Act: north, Rew 0, Score 30, Done False, Value 5.938
Obs: Treasury In times past the vast wealth of the Duke would be stored here. Alas, with the castle taken and
    the Duke gone so with him went his riches. As you remember your history lessons, you recall the
    genealogy of the Dukes of Bostwin and their love of collecting treasures. You could only imagine what
    vast treasures were contained in this room long ago. A finely crafted oaken chest is here.
TmplPred: fwoosh 0.972, south 0.010, north 0.006, put OBJ down 0.004, examine OBJ 0.003
ObjtPred: dagger 0.680, dirty 0.097, sword 0.082, chest 0.080, all 0.043
Act: fwoosh, Rew 10, Score 40, Done False, Value 12.325
Obs: You hear a roar of flames as a fireball materializes out of nowhere and floods the entire room in a burst
     of flames! The intensity of the fireball blasts the chest open and miraculously, except for a slight
     discomfort of warmth, you are otherwise uneffected. [Your score has just gone up by ten points.]
TmplPred: take OBJ 0.822, put on OBJ 0.090, put OBJ in OBJ 0.022, close OBJ 0.019, south 0.015
ObjtPred: jewel 0.904, dagger 0.039, emerald 0.012, small 0.008, sword 0.007
Act: take jewel, Rew 5, Score 45, Done False, Value 5.912
Obs: You feel a surge of power flow through you as you pick up the small emerald...Taken. [Your score has just
     gone up by five points.]
TmplPred: south 0.754, close OBJ 0.089, put OBJ in OBJ 0.066, take OBJ 0.027, put OBJ down 0.024
ObjtPred: dagger 0.441, emerald 0.228, sword 0.186, chest 0.077, floor 0.025
Act: south, Rew 0, Score 45, Done False, Value 1.267
Obs: Armory
TmplPred: east 0.755, examine OBJ 0.096, put OBJ down 0.066, west 0.043, south 0.019
ObjtPred: dagger 0.385, emerald 0.313, floor 0.135, sword 0.118, all 0.031
Act: put dagger down, Rew 0, Score 45, Done False, Value 1.391
Obs: Dropped. The Dark Elf walks in.
TmplPred: enter 0.319, take OBJ 0.294, examine OBJ 0.272, put OBJ down 0.019, west 0.013
ObjtPred: vamvevmew 0.409, dagger 0.182, sword 0.115, emerald 0.079, floor 0.061
Act: push vamvevmew to dagger, Rew 0, Score 45, Done False, Value 3.886
Obs: I don't suppose The Dark Elf would care for that. The Dark Elf senses your aura of power and realizes you
     have the emerald he's been looking for! He bashes you on the back of your head with the hilt of his
     sword and steals the small emerald from you! Your vision swims and you can't quite your bearings... You
     hear alot of shouting nearby between Vamvevmew and a familiar sounding voice followed by an explosion as
     the hair stands up on your head! You draw your sword and attempt to get your bearings but you feel
     displaced... Your vision struggles for clarity as you attempt to look through the smoke and the dust to
     behold two figures locked in sorceral combat! You can make out Morden mumbling something under his
     breath as he gestures towards The Dark Elf and a bolt of lightning hits Vamvevmew squarely in the chest!
     Howling in rage and pain, the Dark One returns the volley with a fire strike which lights up a
     previously invisible defensive shield around Morden! Essentially unhurt, Morden looks weakened from this
     engagement... You look on in bewilderment but know that you must do something! Castle, Upper Level This
     large room has taken on some new charasterics recently... now the walls are covered with blast marks of
     unknown origin from floor to ceiling. Smoke wafts throughout and it's about 10 degrees warmer in here.
     A box made of solid platinum is sitting on the floor here. You can also see a small emerald, The Dark
     Elf and Morden here.
TmplPred: hit OBJ 0.966, undo OBJ 0.010, push OBJ to OBJ 0.004, west 0.004, examine OBJ 0.004
ObjtPred: vamvevmew 0.958, emerald 0.026, dagger 0.005, small 0.003, sword 0.002
Act: hit vamvevmew, Rew 10, Score 55, Done False, Value 9.970
Obs: Wounded by the Wizard Morden and distracted by the mortal combat he is engaged in, you are able to come
     up behind Vamvevmew and run him through with your sword. A look of surprise crosses his face as he looks
     down and realizes he is mortally wounded. He gives you a black look and attempts to reach for your
     throat but doesn't quite make it, crumpling to the floor he breathes his last and dies. [Your score has
     just gone up by ten points.]
TmplPred: undo OBJ 0.205, examine OBJ 0.181, put OBJ on OBJ 0.176, put OBJ down 0.166, push OBJ to OBJ 0.151
ObjtPred: dagger 0.239, floor 0.181, emerald 0.165, box 0.158, all 0.131
Act: undo dagger, Rew 0, Score 55, Done False, Value -0.014
Obs: You can't see any such thing.
TmplPred: undo OBJ 0.207, put OBJ on OBJ 0.174, examine OBJ 0.173, push OBJ to OBJ 0.165, put OBJ down 0.152
ObjtPred: dagger 0.224, floor 0.199, box 0.169, emerald 0.167, sword 0.120
Act: undo box, Rew 0, Score 55, Done False, Value -0.027
Obs: You open the platinum box, revealing The Empirical Emerald.
TmplPred: put OBJ on OBJ 0.174, examine OBJ 0.160, put OBJ down 0.151, push OBJ to OBJ 0.147, put OBJ in OBJ
    0.134
ObjtPred: dagger 0.240, floor 0.196, emerald 0.166, box 0.164, sword 0.117
Act: put box down, Rew 0, Score 55, Done False, Value -0.028
Obs: The platinum box is already here.
TmplPred: put OBJ down 0.171, examine OBJ 0.167, push OBJ to OBJ 0.150, put OBJ on OBJ 0.142, take OBJ 0.119
ObjtPred: dagger 0.229, floor 0.199, box 0.166, emerald 0.165, sword 0.119
Act: close dagger, Rew 0, Score 55, Done False, Value -0.017
Obs: You can't see any such thing.
TmplPred: examine OBJ 0.159, put OBJ down 0.158, put OBJ on OBJ 0.149, push OBJ to OBJ 0.147, take OBJ 0.129
ObjtPred: dagger 0.240, floor 0.178, emerald 0.163, box 0.157, all 0.129
Act: take dagger, Rew 0, Score 55, Done False, Value -0.031
Obs: You can't see any such thing.
TmplPred: push OBJ to OBJ 0.156, examine OBJ 0.150, put OBJ on OBJ 0.148, put OBJ down 0.147, take OBJ 0.134
ObjtPred: dagger 0.237, floor 0.183, emerald 0.164, box 0.159, sword 0.126
Act: put dagger down, Rew 0, Score 55, Done False, Value -0.032
Obs: You can't see any such thing.
TmplPred: put OBJ on OBJ 0.152, push OBJ to OBJ 0.150, put OBJ down 0.148, examine OBJ 0.148, put OBJ in OBJ
    0.139
ObjtPred: dagger 0.236, floor 0.189, emerald 0.164, box 0.159, sword 0.124
Act: put box on floor, Rew 0, Score 55, Done False, Value -0.022
Obs: The platinum box is already here.
TmplPred: examine OBJ 0.161, push OBJ to OBJ 0.161, put OBJ down 0.160, take OBJ 0.136, put OBJ on OBJ 0.134
ObjtPred: dagger 0.233, floor 0.186, box 0.167, emerald 0.158, sword 0.125
Act: take dagger, Rew 0, Score 55, Done False, Value -0.026
Obs: You can't see any such thing.
TmplPred: push OBJ to OBJ 0.170, examine OBJ 0.164, put OBJ down 0.159, put OBJ on OBJ 0.143, take OBJ 0.139
ObjtPred: dagger 0.235, floor 0.181, box 0.165, emerald 0.158, all 0.127
Act: put floor down, Rew 0, Score 55, Done False, Value -0.034
Obs: What do you want to put the floor in?
TmplPred: put OBJ down 0.164, push OBJ to OBJ 0.164, examine OBJ 0.160, put OBJ on OBJ 0.138, take OBJ 0.130
```

```
ObjtPred: dagger 0.234, floor 0.186, box 0.165, emerald 0.158, sword 0.126
Act: put dagger in all, Rew 0, Score 55, Done False, Value -0.021
Obs: You can't see any such thing.
TmplPred: push OBJ to OBJ 0.171, put OBJ on OBJ 0.154, examine OBJ 0.153, put OBJ down 0.149, take OBJ 0.132
ObjtPred: dagger 0.234, floor 0.187, box 0.166, emerald 0.159, sword 0.125
Act: put dagger down, Rew 0, Score 55, Done False, Value -0.029
Obs: You can't see any such thing.
TmplPred: push OBJ to OBJ 0.169, examine OBJ 0.165, put OBJ down 0.160, put OBJ on OBJ 0.152, take OBJ 0.134
ObjtPred: dagger 0.233, floor 0.184, box 0.165, emerald 0.158, sword 0.127
Act: put sword on dagger, Rew 0, Score 55, Done False, Value -0.016
Obs: You can't see any such thing.
TmplPred: push OBJ to OBJ 0.188, examine OBJ 0.167, put OBJ on OBJ 0.162, put OBJ down 0.157, take OBJ 0.141
ObjtPred: dagger 0.227, floor 0.194, box 0.171, emerald 0.160, sword 0.123
Act: put floor on dagger, Rew 0, Score 55, Done True, Value -0.025
Obs: You can't see any such thing.
```

