# OpenReview forum: "Graph Constrained Reinforcement Learning for Natural Language Action Spaces"
_ICLR.cc/2020/Conference — Accept (Poster)_

### Official Review · AnonReviewer1 · 2019-10-23
**Official Blind Review #1**

**Rating:** 6

**Review:**

This paper proposes a knowledge graph advantage actor critic (KG-A2C) model to allow an agent to do reinforcement learning in the interactive fiction game. Under the general framework of A2C, the core contribution of the paper is to apply a graph attention network on the knowledge graph to help learn better representation of the game state and reduce the action space. Experiments on Zork1 game environment are done to verify the effectiveness of the proposed method.

Overall, this paper presents a novel contribution to reinforcement learning with augmented memory/world-state. However, I do have a few concerns regarding the baselines and other details. Given these clarifications and or comparisons in an author response, I would be willing to increase the score.

Pros:

1, I like the idea of constructing the knowledge graph as the agent roll out. I think it is a better way to construct a structural representation of the world rather than assuming the agent gonna learn everything via single hidden state vector. It also permits more explainable policy in the future. Authors do make good progress along this line.

2, The paper is well written and the design of the proposed new model seems technically reasonable.

Cons & Questions:

1, The main concern I had is regarding to the baseline. I think it is more convincing to have a baseline which leverages the same entity extraction and template-action space. In particular, it should have the same model architecture except that it uses maybe a LSTM to decode the action rather than a GAT applied on knowledge graph. Note that the baseline I am referring to is different from the LSTM-A2C baseline reported in the paper as: (1) with entity extraction, although you may not get a graph mask, but you can still have a object-mask which also reduces the action space; (2) it is not clear to me that LSTM-A2C uses the same template-action mechanism as the KG-A2C, e.g., the valid action construction procedure described in section 4.1. Without such a baseline, it is hard to fully judge how helpful the knowledge graph is.

2, How do you test the generalization of the proposed models? In particular, do you use different maps during training and testing? If the model is merely trained on one map as shown in figure 5, it may just memorizes it in the knowledge graph and overfit to this map.

3, The details of the interaction fiction problem setup are sparse. It would be very helpful to explain what exactly the observations are in the example of Figure 2. For example, what are the game description, game feedback are in this case?

4, In the caption of figure 1, “Solid lines represent gradient flow” is misleading. If I understood correctly, solid lines refer to the computation flow which has gradient back-propagated in the backward pass.

5, Could you explain why KG-A2C converge slower than DRRN in figure 3?

6, Do you think having a fully differentiable mechanism of building knowledge graph would help or not? Why?

======================================================================================================

Most of my concerns are addressed by the authors' response. I increased my score.

**Experience Assessment:**

I have published one or two papers in this area.

**Review Assessment: Checking Correctness Of Derivations And Theory:**

I carefully checked the derivations and theory.

**Review Assessment: Checking Correctness Of Experiments:**

I assessed the sensibility of the experiments.

**Review Assessment: Thoroughness In Paper Reading:**

I read the paper at least twice and used my best judgement in assessing the paper.

---

> ### Author Response · Authors · 2019-11-15
> **Reply to Reviewer 1**
>
> We would first like to thank the reviewer for their thoughtful comments and time. We will now address some of the expressed concerns below.
>
> The reviewer expresses concerns regarding the baselines and asks for an ablation without the GAT but with an object mask based on the interactive object entity extraction process. This method reaches an average score of 27.2 on Zork1 when measured in the same manner as the rest of the ablations (over 5 independent runs of 100k steps). We have also updated the paper to add this ablation and a corresponding learning curve in the Appendix. We would like to note that this ablation could be expected to perform worse than the full KG-A2C as it lacks the GAT component in the state representation but also has an advantage as the interactive object mask is less noisy than the graph mask and so presents a smaller action space.
>
> We would also like to clarify here that the LSTM-A2C uses the same decoding process as KG-A2C. That is, a template is first decoded and then objects are filled into the template. The only difference this model has when compared to the KG-A2C is that there are no knowledge graph components.
>
> Regarding the generalization of the proposed models and training/testing on the same maps:
> Our experiments show that use of knowledge graphs is a general technique in the sense that across the various games we see improvements from agents using this method as compared to agents without it. Additionally, the knowledge graph is reset at the beginning of each episode so there is no way for the knowledge graph to memorize the map of the environment. However, we agree with the reviewer that It would be interesting future work to alter the connectivity of games like Zork to assess how well a single agent can generalize across different maps.
>
> To address the request of more details of the interactive fiction problem setup, we would like to direct the reviewer to the transcript presented in Appendix C where the observations are annotated to differentiate description, feedback, etc. We will edit Fig. 2 to more clearly reflect this.
>
> Similarly, we have also edited the caption of Figure 1 to more clearly reflect what the solid lines indicate.
>
> Why does KG-A2C converge slower than DRRN?: KG-A2C is facing a more difficult exploration problem than DRRN. DRRN explores in the space of valid actions. While the size of this space varies by game and number of items in inventory, Zork has approximately 20 valid actions per time step. KG-A2C needs to explore in the space of Templates x Objects. Zork has 237 templates, so even before considering objects we’d expect KG-A2C to require more time to explore this expanded action space. Additionally, KG-A2C’s knowledge graph is empty at the beginning and takes some amount of exploration before it is built up to the point that it effectively aids in further exploration or is able to constrain the choice of objects.
>
> Would a fully differentiable mechanism of building a knowledge graph help?: Yes, we believe that a fully differentiable mechanism for knowledge graph creation would offer more flexibility than our existing OpenIE-rule-based approach. There has been very recent work on this from Zelinka et. al. (https://grlearning.github.io/papers/80.pdf) showing the effectiveness of learning to build a KG representation.

---

### Official Review · AnonReviewer3 · 2019-10-24
**Official Blind Review #3**

**Rating:** 6

**Review:**

This paper tackles the problem of developing agents to solve interactive fiction (IF) games. The authors propose an agent that builds a dynamic knowledge graph of each state from the textual observation provided by the games, while choosing actions from a template-based action space. While both these directions have been explored in prior work as pointed out, this paper combines them effectively to produce an agent that outperfoms existing methods on a benchmark of different IF games.

Pros:
1. Writing is clear, method is easy to understand and design choices are clearly specified.
2. Empirical results are good and presented on real IF games.

Cons:
1. (minor) While the authors test their method on a suite of human-made IF games, it would also be great to have a study on synthetic cases like the Microsoft TextWorld environments, if only to see which aspects of the method are crucial to making the jump from synthetic to real IF games.


Other comments:
1. It looks like the knowledge graph is constructed for every state separately. The DRRN on the other hand incorporates more of the state history. From Figure 3(b), both DRRN and KG-A2C seem to do well - have you analyzed whether these methods are complementary? In general, it would be nice to have some more analysis on all the models across the different games rather than just Zork1.

**Experience Assessment:**

I have published one or two papers in this area.

**Review Assessment: Checking Correctness Of Derivations And Theory:**

N/A

**Review Assessment: Checking Correctness Of Experiments:**

I assessed the sensibility of the experiments.

**Review Assessment: Thoroughness In Paper Reading:**

I read the paper at least twice and used my best judgement in assessing the paper.

---

> ### Author Response · Authors · 2019-11-15
> **Reply to Reviewer 3**
>
> We thank the reviewer for their encouraging and helpful comments.
>
> The reviewer notes that a minor concern of theirs is to test on a synthetic environment such as Microsoft’s TextWorld. In general, we believe that the human made environments are somewhat more diverse in vocabulary, objects, and types of possible interactions than the synthetic environment. We agree that it would be interesting to do a more thorough comparison between synthetic textworld environments and human-made ones.
>
> The second concern was regarding how DRRN incorporates more state history and whether or not DRRN and KG-A2C were complementary. We would like to clarify that the knowledge graph is constructed incrementally throughout an episode. In particular, the observation at each step produces an update to the knowledge graph, and information is accumulated until the end of the episode. Thus KG-A2C actually retains quite a more state history than DRRN. It may be possible to combine the choice-based action-selection of DRRN with the knowledge graphs of KG-A2C, but this is left to future work.
>
> In the spirit of adding more analysis, we updated the paper to feature a baseline suggested by Reviewer 1. This baseline uses the interactive objects entity extraction feature to provide a less noisy mask which is used to constrain generation instead of the knowledge graph. Other knowledge components such as the GAT in the state embedding are also removed to help assess their effect on the overall architecture.

---

### Official Review · AnonReviewer2 · 2019-10-29
**Official Blind Review #2**

**Rating:** 6

**Review:**

This paper considers the problem of interactive fiction games in which an agent interacts with the world purely through natural language. The problem is challenging because one has to map natural language as observations to appropriate representations to partially infer the state space of the world, and actions are also defined in vast space of natural language sentences. One of the main contribution in the paper is to represent the partial observations of the world as a knowledge graph, and so as to efficiently infer appropriate actions.

The paper is very well written, especially the introduction section, demonstrating novelty in the context of fictional games literature, and showing good empirical results.

However, I don't have any background in fictional games but dialog modeling. So it is hard for me to fairly assess how novel this work is. Ideas are simple and incremental, even if i rely upon literature overview provided by the authors in the related work section. Though, it can be effective.

The problem is challenging, and yet narrows down to fictional games. The proposed solution doesn't seem generic to be applied in other NLP or ML problems.

Authors argue that action space is super large even if generating sentences of length upto 5. Even though true, I think, this argument doesn't hold in the context of recent progress in NLP for problems like dialog modeling, where the action space of generating responses is even larger. I suggest the authors to relate their work to dialog models, as some of the ideas can be borrowed from there to simplify the solution for the considered problem setting.


**Experience Assessment:**

I have published one or two papers in this area.

**Review Assessment: Checking Correctness Of Derivations And Theory:**

I assessed the sensibility of the derivations and theory.

**Review Assessment: Checking Correctness Of Experiments:**

I assessed the sensibility of the experiments.

**Review Assessment: Thoroughness In Paper Reading:**

I read the paper at least twice and used my best judgement in assessing the paper.

---

> ### Author Response · Authors · 2019-11-15
> **Reply to Reviewer 2**
>
> We thank the reviewer for their effort towards improving our manuscript.
>
> To clarify the novelty of our approach - we are the first to explore the ideas of using knowledge graphs specifically to constrain language generation in the context of human-made Interactive Fiction games. Prior work has shown the utility of using knowledge graphs on synthetic TextWorld games, but always in the context of selecting from a small number of discrete actions. In contrast we apply similar techniques to the more challenging domains of human-made games with the far more challenging aspect template-based action generation. The novel insights which made this work possible was the dual uses of the knowledge graph to both represent world state as input to the agent, and also to constrain the generation of actions.
>
> In terms of the difficulty of language generation: Broadly, we agree with the reviewer that modern NLP models generate compelling natural language over far larger spans than the 4-5 words required for Interactive Fiction games. However, these models also benefit from supervised training on vast amounts of text. In contrast, IF agents like KG-A2C start learning tabula rasa without the benefit of supervised labels indicating the correct actions to generate. We are unaware of any dialog modeling systems that are trained tabula-rasa from temporally delayed rewards. We would be receptive to learning of any techniques or ideas from dialog modeling which may be applicable to the setting of text-based games.
>
> The reviewer expresses concerns regarding the generalizability of the proposed solution to other ML/NLP problems. We agree that the techniques we developed for Interactive Fiction games are not tested in other settings. However, we are optimistic that there is some commonality between the types of challenges faced by an RL agent in a text-game and a task-oriented dialog-system that is interacting with a user. In particular, both need to reason about the latent state of the environment / dialog and choose contextually relevant actions to perform. Similarly, both need to compactly represent that history of past observations / utterances in order to reason about what action to execute. We believe the technique of using a knowledge graph to efficiently capture relevant aspects of past observations may also be applicable to dialog settings to capture information from past interactions. Additionally, the use of the knowledge stored in the graph to constrain action selection is a technique that may transfer well to other settings.

---

### Decision · Program_Chairs · 2019-12-19

**Decision:**

Accept (Poster)

**Comment:**

This paper applies reinforcement learning to text adventure games by using knowledge graphs to constrain the action space. This is an exciting problem with relatively little work performed on it. Reviews agree that this is an interesting paper, well written, with good results. There are some concerns about novelty but general agreement that the paper should be accepted. I therefore recommend acceptance.